health and disease and epidemiology/theoretical biology/mathematical modelling

stochastic process, epidemic models, Monte Carlo, fluctuations

**Author for correspondence:**
Alexandre S. Martinez
e-mail: asmartinez@usp.br

# Improved susceptible–infectious–susceptible epidemic equations based on uncertainties and autocorrelation functions

Gilberto M. Nakamura[1,2,3], George C. Cardoso[2] and Alexandre S. Martinez[2,3]

[1]Université Paris-Saclay, CNRS/IN2P3, and Université de Paris, IJCLab, 91405 Orsay, France
[2]Faculdade de Filosofia, Ciências e Letras de Ribeirão Preto (FFCLRP), Universidade de São Paulo (USP), Ribeirão Preto 14040-901, Brazil
[3]Instituto Nacional de Ciência e Tecnologia – Sistemas Complexos (INCT-SC), Rio de Janeiro, Brazil

GMN, 0000-0001-7803-1331; GCC, 0000-0001-8459-9812; ASM, 0000-0002-4395-0511

Compartmental equations are primary tools in the study of disease spreading processes. They provide accurate predictions for large populations but poor results whenever the integer nature of the number of agents is evident. In the latter instance, uncertainties are relevant factors for pathogen transmission. Starting from the agent-based approach, we investigate the role of uncertainties and autocorrelation functions in the susceptible–infectious–susceptible (SIS) epidemic model, including their relationship with epidemiological variables. We find new differential equations that take uncertainties into account. The findings provide improved equations, offering new insights on disease spreading processes.

## 1. Introduction

Communicable diseases are health disorders caused by pathogens transmitted from infected individuals to susceptible ones [1]. In general, the transmission process occurs with variable success rate, subjected to stochastic uncertainties during the infectious period of the host. These uncertainties comprehend aspects related to biological transmission mechanisms and availability of adequate contact between hosts and susceptible individuals. For large and well-connected populations, stochastic factors are discarded in favour of deterministic differential equations, also known as compartmental or mean-field equations [2–4]. Recent advances in network theory [5] provided a far more clear picture of interactions among elements of

the population, improving predictions for heterogeneous social structures. Generalizations for compartmental equations have been able to reproduce pandemics and prove analytical results, taking into account complex network topologies, highlighting the role of central hubs in general disease spreading dynamics [6–11].

By contrast, the stochastic nature of disease transmission cannot be omitted for a number of scenarios. It becomes more pronounced for small populations, where the characteristics of each agent forming the population are relevant variables to the spreading process [12]. Incidentally, this is often the case in emerging diseases [13]. Because the population cannot be treated as homogeneous, average values are no longer adequate, impacting the accuracy of compartmental equations. Stochastic models deal with the issue by proposing simpler rules to express the disease transmission, taking the relevant stochastic factors into account. More importantly, the stochastic analysis expands the machinery used to study the problem beyond population averages. It includes tools such as correlations [12] and autocorrelation functions, which extract inner details of the stochastic dynamics and subsequently provide insights to solve them. For instance, in the standard Brownian motion, the autocorrelation function of the position displays a delta-like behaviour due to white noise, i.e. $\langle x(t)x(t')\rangle \propto \delta(t-t')$. This means that the position of a particle $x(t)$ is uncorrelated to its position $x(t')$ at time $t'$, except when $t=t'$. This feature leads to the well-known linear growth of the spatial variance with time [14]. In disease spreading, autocorrelation functions have also been used to study time series of epidemiological data and assess the impact of spatial influences on stochastic fluctuations [15–20].

Here, we derive exact differential equations for both the instantaneous average density of infected agents, $\langle \rho(t)\rangle$, and its corresponding variance, $\sigma^2(t)$, in the susceptible–infectious–susceptible (SIS) epidemic model with $N$ agents. We find that uncertainties play an important role in small populations or small prevalence of the disease, impacting estimates of epidemiological parameters from data. Numerical and analytical evidence allow us to formulate two closure relations for $\langle \rho^3(t)\rangle$, and derive systems of differential equations for $\langle \rho(t)\rangle$ and $\sigma^2(t)$. The selection of the appropriate closure relation depends solely on the nature of the fluctuations present in the system. This issue has been examined in detail before [21–24]. It turns out that the nature of the fluctuations can be assessed from the normalized autocorrelation function $D_{\rho\rho}(t)$, including scenarios with finite population sizes. Non-Gaussian fluctuations develop whenever the absorbing state (disease eradication) influences the outcome of disease spreading [21]. We exploit the relationship between $D_{\rho\rho}(t)$ and $\langle \rho^3(t)\rangle$ to craft a closure relation in this case. For non-Gaussian fluctuations, a different closure relation emerges as a consequence of vanishing skewness coefficient $\kappa_3(t)$. The resulting differential equations for Gaussian fluctuations have been reported before [25,26], and also derived in a more general formulation for population dynamics based on Langevin equations [27]. We combine the system of equations into a single nonlinear second-order differential equation, and discuss an analytical solution. The new equations provide significant improvements over the traditional compartmental equation, as they account for stochastic effects, while being far more amenable to analytical studies than the master equation of the disease spreading process.

This paper is organized as follows. Section 2 opens our discussion with compartmental equations of the SIS model, with emphasis on general aspects of parameter estimation. Section 3 reviews the spreading process under the agent-based approach. Improved differential equations for the SIS model are derived. Analytical and numerical properties of $D_{\rho\rho}(t)$ are investigated in §4, leading to dynamics for non-Gaussian fluctuations. Dynamics for Gaussian fluctuations are addressed in §5. In each case, the systems of differential equations are combined producing two distinct second-order differential equations for $\langle \rho(t)\rangle$. In §6, we present our closing arguments, remarks and potential applications.

## 2. Compartmental equations

Let $\rho(t)$ be the density of infected agents in a population of size $N$ in the SIS model. In the compartmental approach, the population is assumed to be large, homogeneous and highly interconnected. As a result, agents can be regarded as statistically equivalent, and $\rho(t)$ becomes a good descriptor of the system. The assumptions impose that the system must be, on average, invariant under permutations. The simplest way to satisfy permutation invariance assumes agents connected to each other. Incidentally, this population structure shares the same characteristics as the complete graph [4,28].

The other relevant assumption concerns the transmission mechanism. Because the population is taken as homogeneous, the adequate interaction between infected and susceptible agents occurs with probability proportional to $(1-\rho)\rho$. This assumption constitutes the basis for the random mixing hypothesis [3]. At the same time, recovery events are proportional to the infected density $\rho$. Following this notation, the SIS compartmental equation for $\rho(t)$ reads

$$\frac{d\rho}{dt} = \alpha(1-\rho)\rho - \gamma\rho, \tag{2.1}$$

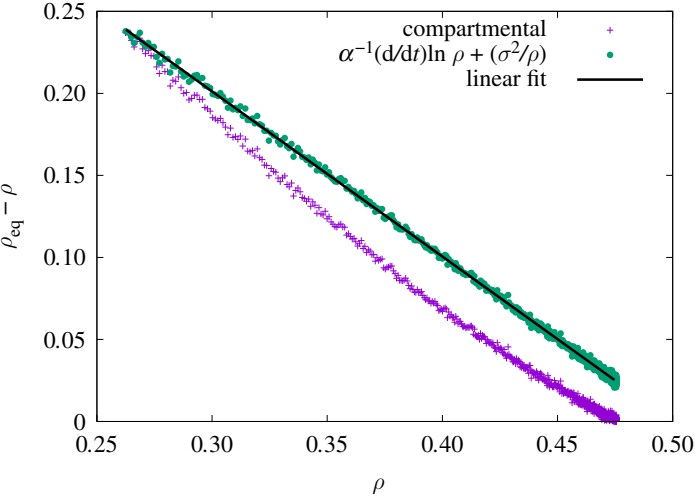

**Figure 1.** Deviations from compartmental predictions. Predicted values of $\rho_{eq} - \rho$ versus observed $\rho$ with (full circles) and without (cross) corrections. Corrections are related to $\sigma^2/\rho$, where $\sigma^2$ is the variance of $\rho$. Monte Carlo simulations are performed with $10^6$ samples in the complete graph with $N = 50$ agents, $\gamma = 1/2$ and $\alpha = 1$. Linear fit (solid line) produces $\gamma_{data} = 0.50(3)$ and $\alpha_{data} = 1.00(0)$.

where $\alpha$ and $\gamma$ are the transmission and recovery rate, respectively. Explicit generalizations are available for several different networks [6–8], including complex networks. These special network structures highlight the role of super-spreaders in real-world spreading processes [3].

For data fitting and parameter estimation purposes, it is convenient to consider the relative variation of $\rho(t)$ over time. Rearranging equation (2.1) and defining the steady-state density $\rho_{eq} = 1 - \gamma/\alpha$, we obtain

$$\frac{1}{\rho}\frac{d\rho}{dt} = \frac{d}{dt}\ln\rho = \alpha(\rho_{eq} - \rho). \tag{2.2}$$

From epidemiological data, equation (2.2) provides a simple way to extract $\alpha$ and $\gamma$ by a linear fit. For $\alpha \geq \gamma$, dividing equation (2.2) by $\rho(t)$ and plugging the solution $\rho(t) = \rho_{eq}/(1 - C_1 e^{-\rho_{eq}\alpha t})$, with $C_1 = 1 - (\rho_{eq}/\rho(0))$, leads to a simple exponential decay

$$\frac{1}{\rho}\frac{d}{dt}\ln\rho = \alpha\left[\frac{\rho_{eq}}{\rho(0)} - 1\right]e^{-\alpha\rho_{eq}t}, \tag{2.3}$$

where the decay rate depends only on epidemiological parameters. Again, equation (2.3) can be used to extract $\alpha$ and $\gamma$ using a linear fit in logscale.

It should be clear by now that equation (2.2) is an important tool to extract epidemiological parameters. What would be the implications for epidemiological studies if equation (2.2) had additional terms or corrections? Figure 1 displays the values of $\alpha^{-1}(d/dt)\ln\rho$ using equation (2.2) with data from numerical simulations (see Data accessibility and [29] for further details). In this controlled computational experiment, predictions for $\alpha^{-1}(d/dt)\ln\rho$ deviate from $\rho_{eq} - \rho$. Even more, figure 1 shows that early estimates of epidemiological parameters, typical during the onset of epidemics, underestimate the transmission rate. Since agents are equivalent to each other in this setting, the only remaining source of error is due to the discrete nature of transmission and recovery events. Therefore, inherent stochastic events in spreading processes affect predictions whenever uncertainties cannot be neglected. Thus, it seems reasonable to examine more closely this limitation of the compartmental equations, which are far more familiar to epidemiology practitioners [30].

In what follows, we calculate corrections for equation (2.2) using a stochastic agent-based approach to better grasp the emergence of uncertainties in the SIS model. The approach allows for a direct comparison with numerical simulations, and it does not require the coupling of dynamical equations with an external noise source to mimic fluctuations (Langevin formulation). We also note that the averaging procedure employed in the numerical simulations is equivalent to the ensemble averaging. In an ensemble average, the averages are estimated from a large set of independent realizations of a given stochastic process that share the same initial conditions. One way to create an approximate ensemble from real epidemiological data consists in partitioning the system into smaller subsets that are weakly interacting with each other. This is the basis of ensemble formation in physics [31] and the

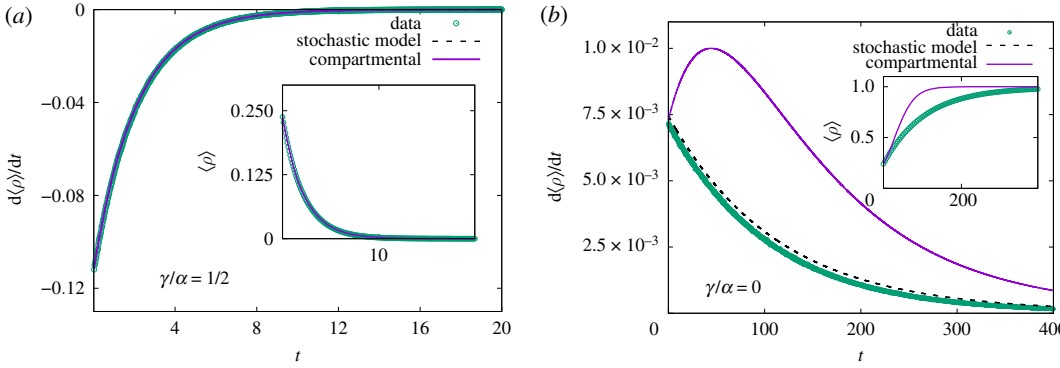

**Figure 2.** Linear chain. Simulated data for SIS agent-based model with $N = 50$ agents, in a linear chain with periodic boundary condition. The system features translation symmetry but its low connectivity violates the random-mixing hypothesis. (*a*) Simulated data agree with predictions obtained from compartmental equations as recovery events dominate the dynamics. (*b*) Diffusion of the disease in the linear chain creates correlations between agents. The operator formalism and translation symmetry (dashed line) provide an improved prediction for variation rate of the average density of infected, $\mathrm{d}\langle\rho\rangle/\mathrm{d}t = (2\alpha/N)[\langle\rho\rangle - (1/N)\sum_k\langle n_k n_{k+1}\rangle] - \gamma\langle\rho\rangle$.

core hypothesis of metapopulation models [32]. Even so, this is only a coarse representation of the idealized ensemble. The problem is somewhat reduced in numerical simulations as the number of realizations can be increased in exchange for computing time. In short, the advantage of ensemble averaging is that it makes it possible to find equations that describe the general behaviour of stochastic variables—for instance, the diffusion equation for the random walker.

# 3. Stochastic formalism

In the agent-based approach [33,34], the population consists of $N$ distinguishable agents connected to each other according to a predefined adjacency matrix $A$ ($N \times N$). Each agent ($k = 0, 1, \cdots, N-1$) may assume one of two possible health states $n_k$ in the SIS model, either susceptible ($n_k = 0$) or infected ($n_k = 1$). Following [35,36], there are $2^N$ available configurations in the canonical basis $|\mu\rangle$, with $\mu = 0, 1, \cdots, 2^{N-1}$. Configurations are readily extracted from the binary construction $\mu = n_0 2^0 + n_1 2^1 + \cdots + n_{N-1} 2^{N-1}$. As an example, for $N = 4$, the configuration $|0\rangle = |0\,0\,0\,0\rangle$ represents the infected-free configuration, whereas all agents are infected in $|15\rangle = |1\,1\,1\,1\rangle$.

Here, we treat the disease spreading process as a Markov process. The corresponding master equation reads

$$\frac{\mathrm{d}}{\mathrm{d}t}|P(t)\rangle = -\hat{H}|P(t)\rangle, \tag{3.1}$$

where $|P(t)\rangle = \sum_{\mu=0}^{2^N-1} P_\mu(t)|\mu\rangle$ is the probability vector, with $P_\mu(t)$ being the instantaneous probability to find the system in the configuration $|\mu\rangle$; and $\hat{H}$ is the generator of time translations, given by the following expression:

$$\hat{H} = \frac{\alpha}{N}\sum_{k,\ell=0}^{N-1} A_{k\ell}(1 - \hat{n}_k - \hat{\sigma}_k^+)\hat{n}_\ell + \gamma\sum_{k=0}^{N-1}(\hat{n}_k - \hat{\sigma}_k^-). \tag{3.2}$$

Operators are assigned the hat symbol to distinguish them from scalars. The operators $\hat{n}_k$ extract the health state of the $k$-th agent, $\hat{n}_k|n_0\cdots n_k\cdots\rangle = n_k|n_0\cdots n_k\cdots\rangle$, while $\hat{\sigma}_k^\pm$ are the usual spin-1/2 ladder operators, i.e. $\hat{\sigma}_k^+|n_0\cdots 0_k\cdots\rangle = |n_0\cdots 1_k\cdots\rangle$ and $\hat{\sigma}_k^-|n_0\cdots 1_k\cdots\rangle = |n_0\cdots 0_k\cdots\rangle$, respectively. The main advantage of using equations (3.1) and (3.2) lies in their applicability for arbitrary networks, without further assumptions on the probability distribution.

As an example, consider a system in which the adjacency matrix describes a linear chain with periodic boundary conditions, i.e. $A_{k\ell} = \delta_{k,\ell\pm1}$ and $A_{0,N-1} = A_{N-1,0} = 1$. Clearly, the connections between agents are invariant under translations, so that agents are statistically equivalent. However, the number of connections is now reduced to two instead of $N-1$ and violates the random mixing hypothesis. Figure 2 exhibits numerical simulations for the linear chain and predictions using the compartmental equation with effective transmission rate $(2/N)\alpha$. It should come as no surprise that the predictions become increasingly worse for vanishing $\gamma$, since agents can only infect their nearest neighbours, thus

introducing correlations. By contrast, the agreement between simulated data and the prediction provided by equations (3.1) and (3.2) is far more accurate, reinforcing their validity for general networks.

Despite the known effects of network structures on the dynamics of epidemics [11,28,37,38], there are instances in which the uniqueness of agents can be a minor concern. In these cases, uncertainties stem from the stochastic nature of disease spreading processes. They produce additional corrections to the dynamical equations, with enhanced effects for a finite population of size $N$. We set aside the complexities associated with network structures by adopting the complete graph. The complete graph replicates the random-mixing hypothesis because each agent interacts with the remaining $N-1$ agents, $A_{ij} = 1 - \delta_{ij}$. The choice also allows an adequate comparison with the compartmental equations.

Equations (3.1) and (3.2) can be used to evaluate statistics relevant to the epidemic model. Among them, the average density of infected agents,

$$\langle \rho(t) \rangle \equiv \frac{1}{N} \sum_{\mu=0}^{2^N-1} \eta_\mu P_\mu(t), \tag{3.3}$$

where $\eta_\mu \equiv \sum_k \langle \mu | \hat{n}_k | \mu \rangle$ is the total number of infected agents in the configuration $|\mu\rangle$. By virtue of equation (3.3), it is clear that the time derivative of $\langle \rho(t) \rangle$ depends solely on $\mathrm{d}P_\mu/\mathrm{d}t$. In turn, equation (3.1) states $\mathrm{d}P_\mu/\mathrm{d}t = -\sum_\nu \langle \mu | \hat{H} | \nu \rangle P_\nu = -\sum_\nu H_{\mu\nu} P_\nu$, which concerns the calculation of the matrix elements $H_{\mu\nu}$. Although their explicit evaluation exists, we are actually interested in the summation $\sum_\mu \eta_\mu H_{\mu\nu}$. The latter can be easily calculated noting that $\sum_\mu \eta_\mu \langle \mu | \sum_k \hat{\sigma}_k^+ | \nu \rangle = (\eta_\nu + 1)(N - \eta_\nu)$ and $\sum_\mu \eta_\mu \langle \mu | \sum_k \hat{\sigma}_k^- | \nu \rangle = (\eta_\nu - 1)\eta_\nu$. More specifically, the non-vanishing matrix elements $\langle \mu | \sum_k \hat{\sigma}_k^+ | \nu \rangle$ connect configurations whose number of infected agents differ by one, and the number of possible configurations is $N - \eta_\nu$. For example, with $N = 4$ and $|\nu\rangle = |0010\rangle$, the configurations in question are $|1010\rangle$, $|0110\rangle$, $|0011\rangle$. A similar argument can be made for $\sum_k \hat{\sigma}_k^-$, reducing $\eta_\mu$ by one and $\eta_\nu$ matching configurations. Therefore,

$$\frac{\mathrm{d}\langle \rho \rangle}{\mathrm{d}t} = -\frac{1}{N} \sum_{\mu=0}^{2^N-1} \eta_\mu H_{\mu\nu} P_\nu(t) = \alpha[\rho_{\mathrm{eq}} - \langle \rho(t) \rangle]\langle \rho(t) \rangle - \alpha\sigma^2(t), \tag{3.4}$$

with instantaneous variance $\sigma^2(t) = \langle \rho^2 \rangle - \langle \rho \rangle^2$ (see appendix A for details). A brief inspection of equation (3.4) shows that the correction $-\alpha\sigma^2(t)$ always slows down the growth rate of $\langle \rho(t) \rangle$. As a result, it directly affects the estimation of epidemiological parameters as shown in figure 1.

We emphasize that the inherent fluctuations of the disease spreading process is summarized by $\sigma^2(t)$ in equation (3.4). An initial uncertainty evolves during the course of the spreading process, restricted by the fact that agents can only be either susceptible or infected, i.e. there is no half infection nor half cure. In a sense, $\sigma^2(t)$ shares the concept of shot noise in condensed matter physics [39]. Moreover, equation (3.4) recovers equation (2.1) for vanishing $\sigma^2(t)$, a situation that often arises for large populations since the relative uncertainty scales with $N^{-1/2}$. For small populations or small values of $\langle \rho(t) \rangle$, equation (3.4) highlights the influence of noise in the spreading process, even if agents are statistically equivalent.

Noting that $\sigma^2(t)$ depends on time, there must exist an additional differential equation for $\sigma^2(t)$. Indeed, the same rationale behind equation (3.4) can be used to find $(\mathrm{d}/\mathrm{d}t)\sigma^2$, as detailed in appendix A ( in accordance with [26] or [27]). The equation of motion reads

$$\frac{1}{2}\frac{\mathrm{d}\sigma^2}{\mathrm{d}t} = \alpha[\rho_{\mathrm{eq}} + \langle \rho \rangle]\sigma^2 - \alpha\Delta_3(t) + \frac{\alpha}{2N}\langle \rho(1-\rho) \rangle + \frac{\gamma}{2N}\langle \rho \rangle, \tag{3.5}$$

where $\Delta_3(t) = \langle \rho^3(t) \rangle - \langle \rho(t) \rangle^3$. Again, numerical simulations support our findings (see figures 3 and 4). Despite the encouraging results, equation (3.5) creates an explicit dependence on the third statistical moment $\langle \rho^3(t) \rangle$, even if $o(1/N)$ corrections are omitted. Formally, we could calculate the differential equation for $\Delta_3(t)$ but then we would have to deal with $\langle \rho^4(t) \rangle$ and so on, creating a set of hierarchic equations for the statistical moments of $\rho(t)$.

Let us briefly assume that it is possible to estimate a surrogate dynamic for $\Delta_3(t)$ by some ingenious method. In this case, equations (3.4) and (3.5) form a system of differential equations for $\langle \rho(t) \rangle$ and $\sigma^2(t)$. However, $\Delta_3(t)$ also measures the fluctuation strength and it can change radically for different sets of parameters $(N,\gamma/\alpha)$ as long as $N$ remains finite. The inset in figure 4a exhibits the changes in $\sigma^2(t)$ as one reduces $N$. Holding $N$ fixed and varying $\gamma/\alpha$ also triggers this phenomenon. Figure 5 provides a concrete example of two distinct behaviours for fluctuations for fixed $N$: Gaussian and non-Gaussian fluctuations. An existing relationship between $\Delta_3(t)$ and the instantaneous coefficient of skewness,

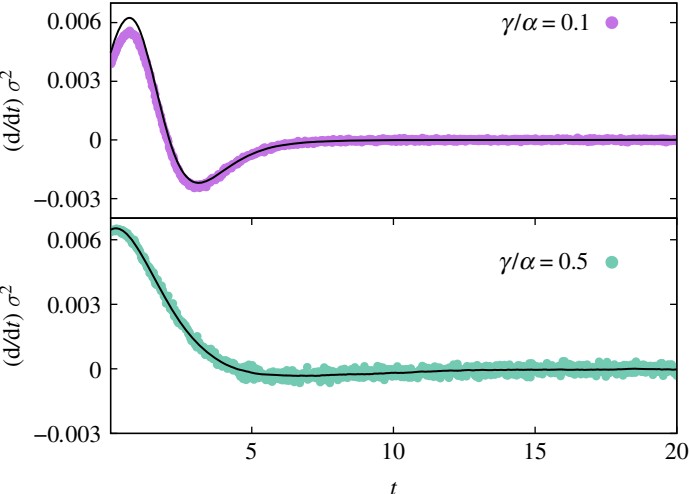

**Figure 3.** Rate of change for the variance in agent-based simulations in finite populations. Simulations are performed over $10^6$ Monte Carlo samples and $N = 50$ agents. Forward time derivative of $\sigma^2(t)$ using simulated data (circles), with $\gamma/\alpha = 0.1$ and 0.5. The solid line represents equation (3.5).

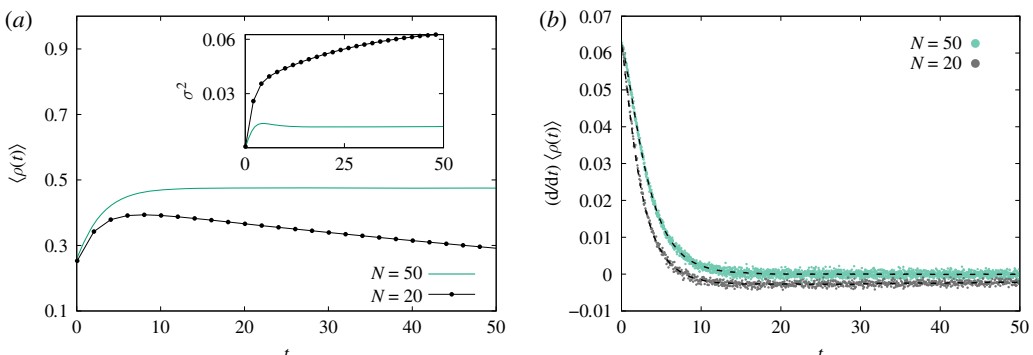

**Figure 4.** Finite size effects in the complete graph with $\gamma/\alpha = 1/2$. (a) For $N = 20$ (dotted lines), the influence of absorbing state drives $\langle\rho\rangle$ below the expected $\rho_{eq} = 1/2$, while $\sigma^2(t)$ increases over time (inset). For $N = 50$ (solid line), $\langle\rho\rangle$ lies slightly below $\rho_{eq}$, with constant $\sigma^2$ for large $t$. (b) Both cases are in agreement with equation (3.4). Simulated data with $10^6$ samples under the same initial condition.

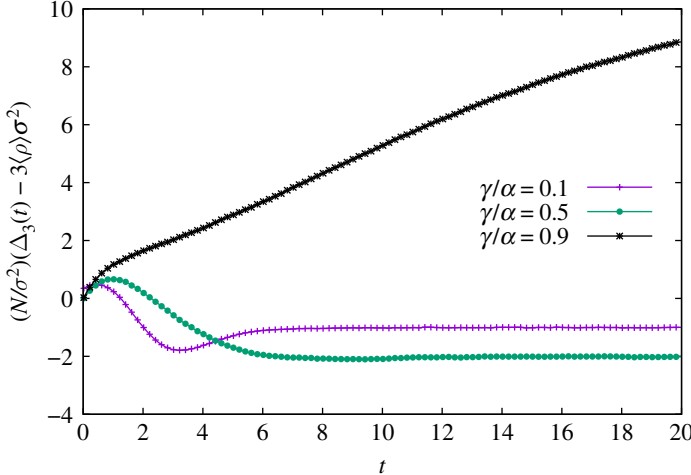

**Figure 5.** Deviations from Gaussian behaviour. Simulations are performed in the complete graph with $N = 50$ agents, and $10^6$ samples. The quantity $\Delta_3 - 3\langle\rho\rangle\sigma^2$ measures the deviation of the system compared to Gaussian fluctuations. Curves for $\gamma/\alpha = 0.1$ and 0.5 imply $\Delta - 3\langle\rho\rangle\sigma^2 \sim o(\sigma^2/N)$. This behaviour is not observed for $\gamma/\alpha = 0.9$, suggesting that the variance vanishes more rapidly than $\Delta_3 - 3\langle\rho\rangle\sigma^2$, in disagreement with Gaussian behaviour. Error bars omitted.

$\kappa_3(t)$, provides a way to investigate symmetric fluctuations [26]. Likewise, the density autocorrelation function provides insights on $\Delta_3(t)$ for non-symmetric fluctuations. Since the nature of these two types of fluctuations is so dissimilar, we shall study them separately.

# 4. Autocorrelation function

Uncertainties described by Gaussian fluctuations are expected to play a significant role in widespread epidemics. However, the situation changes when a small fraction of the agents is infected. The SIS model used in this paper does not account for external infection sources, such as wild animals or immigration; once the number of infected vanishes the spreading process comes to a halt. This constraint means that the absorbing state $|0\rangle$ prevents the occurrence of symmetric probability distributions around low densities. The effect can be found in large populations but it is enhanced in small populations: fluctuations can eradicate the disease. Thus, we need to look for a statistics other than $\kappa_3(t)$ to model the dynamics of $\Delta_3(t)$ for non-Gaussian fluctuations. The statistics should involve, at most, $\rho(t)$ up to the power two; otherwise, it could reintroduce higher statistical moments. In that regard, two-point autocorrelation functions fulfil these requirements.

Let $C_{\rho\rho}(t)$ be the instantaneous autocorrelation function between $\rho(t)$ and $\rho(t + \delta t)$, lagged by a single time window

$$C_{\rho\rho}(t) \equiv \langle \rho(t + \delta t)\rho(t)\rangle - \langle \rho(t)\rangle^2. \tag{4.1}$$

Here, averages are evaluated by considering samples from an ensemble instead of the usual Fourier transform, as the ergodic hypothesis is unavailable. For Markov processes,

$$\langle \rho(t + \delta t)\rho(t)\rangle = \frac{1}{N^2} \sum_\mu \sum_{k,j} \langle \mu | \hat{n}_k e^{-\hat{H}\delta t} \hat{n}_j | P(t)\rangle. \tag{4.2}$$

The evaluation of this expression involves the same rationale used for equation (3.4), as detailed in appendix B. Plugging the result into equation (4.1), we find $C_{\rho\rho}(t) = \sigma^2(t) + \alpha\delta t[\rho_{eq}\langle\rho^2\rangle - \langle\rho^3\rangle] + o(\delta t^2)$. The crucial information here is the relationship between $\langle\rho^3(t)\rangle$ and $C_{\rho\rho}(t)$: provided $C_{\rho\rho}(t)$ can be fitted from epidemiological data, it seems plausible to use it to model $\langle\rho^3(t)\rangle$ and, thus, create a surrogate dynamics for $\Delta_3(t)$. Unfortunately, the lack of a simple functional form prevents the fitting of $C_{\rho\rho}(t)$ with at most two parameters.

Instead, consider the normalized autocorrelation function

$$D_{\rho\rho}(t) \equiv \frac{C_{\rho\rho}(t) - \sigma^2(t)}{\alpha\delta t\langle\rho\rangle^2} = \rho_{eq} - \frac{\langle\rho^3\rangle}{\langle\rho\rangle^2} + \rho_{eq}\frac{\sigma^2}{\langle\rho\rangle^2}. \tag{4.3}$$

For vanishing $\sigma^2(t)$ and $N \gg 1$, $D_{\rho\rho}(t) \approx \rho_{eq} - \langle\rho(t)\rangle$ recovers the r.h.s. of equation (2.2). Hence, $D_{\rho\rho}(t)$ can be associated with $(\mathrm{d}/\mathrm{d}t)\ln\langle\rho\rangle$ in the same limit.

According to equation (2.3), an exponential decay of $D_{\rho\rho}(t)/\langle\rho(t)\rangle$ occurs whenever $\langle\rho(t)\rangle$ is reasonably described by compartmental equations. As the system evolves, $D_{\rho\rho}(t)/\langle\rho(t)\rangle$ experiences a strong divergence (figure 6). Afterwards, $D_{\rho\rho}(t)/\langle\rho(t)\rangle$ either converges to a constant value; or engages in a regime of exponential growth (figure 7). The first case signals that $\langle\rho(t)\rangle$ describes the spreading process with uncertainties summarized by $\sigma^2(t)$. Fluctuations that increase $\rho(t)$ are as likely as those that decrease it. Thus, the probability density function (pdf) associated with of the fluctuations of $\langle\rho(t)\rangle$ is symmetrical. We call them Gaussian fluctuations for the lack of a better name.

By contrast, an exponential growth of $D_{\rho\rho}(t)/\langle\rho(t)\rangle$ exposes the influence of the absorbing state on the evolution of the system. Its impact becomes more noticeable as $\langle\rho(t)\rangle$ approaches the disease eradication, and for small population sizes. In such cases, the fluctuation pdf becomes asymmetrical, resulting in the degradation of $\langle\rho(t)\rangle$ in contradiction with equation (2.1). The fluctuations, in this case, are non-Gaussian. Therefore, $D_{\rho\rho}/\langle\rho\rangle$ separates fluctuations into two distinct classes: Gaussian and non-Gaussian.

One could argue that a reciprocal timescale $\tau^{-1} = (1/2)(\mathrm{d}/\mathrm{d}t)\ln(D_{\rho\rho}^2/\langle\rho\rangle^2)$ emerges because the exponential decay becomes the dominant mode of $\langle\rho(t)\rangle \propto \rho_1 e^{-t/\xi}$, after some time instant $t$, with peak value $\rho_1$. Table 1 exhibits a few estimates for $\tau^{-1}$ and $\xi^{-1}$ from which one can infer $\tau = \xi/2$. As a result, $D_{\rho\rho}(t) \propto e^{t/\lambda}$ with $\lambda = \xi$ after non-Gaussian fluctuations are in place.

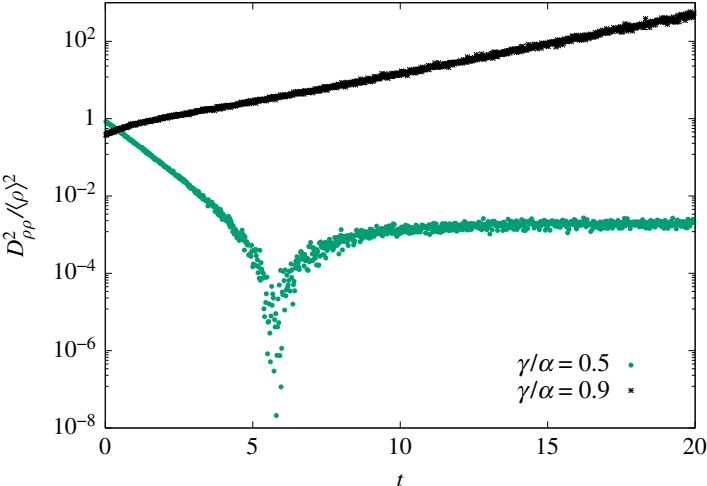

**Figure 6.** Contributions for $|D_{\rho\rho}(t)/\langle\rho\rangle|^2$. Simulation results comprehend $10^6$ simulation samples in the complete graph with $N = 50$. Gaussian fluctuations occur for $\gamma/\alpha = 0.5$ (green circles). An exponential decay is observed during the transient. The divergence appears as $\langle\rho\rangle$ approaches $\rho_{eq}$. Finite size corrections drive $\langle\rho(\infty)\rangle$ to slightly lower values than $\rho_{eq}$ in the steady state. Non-Gaussian fluctuations create an exponential growth during the transient regime for $\gamma/\alpha = 0.9$ (black asterisk).

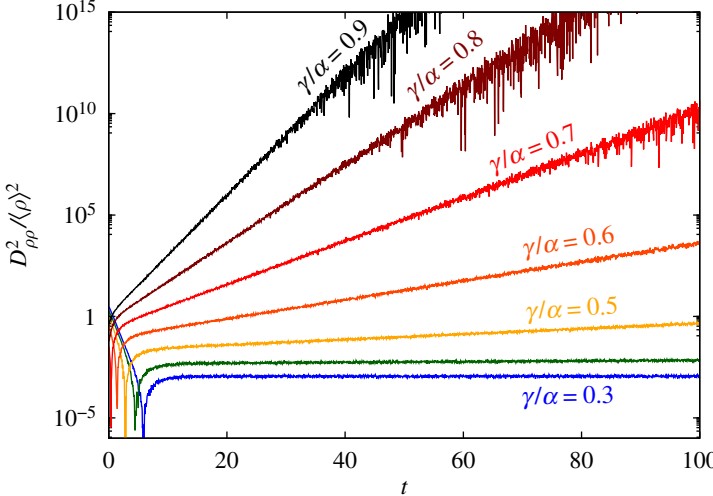

**Figure 7.** $D_{\rho\rho}^2(t)/\langle\rho\rangle^2$ for various ratios $\gamma/\alpha$. Data extracted from numerical simulations with $N = 20$ agents ($10^6$ samples). After a sharp divergence, $|D_{\rho\rho}(t)/\langle\rho\rangle|^2$ either moves towards a constant value (two lowermost curves, $\gamma/\alpha = 0.3$ and $0.4$) or increases exponentially.

**Table 1.** Reciprocal times derived from simulated data with $N = 20$ agents. $|D_{\rho\rho}/\langle\rho\rangle| \propto e^{t/\tau}$, $\langle\rho\rangle \propto e^{-t/\xi}$, and $D_{\rho\rho}(t) \propto e^{t/\lambda}$. Values are consistent with $\tau = \xi/2$ and $\lambda = \xi$.

| $\gamma/\alpha$ | $\tau^{-1}$ | $\xi^{-1}$ | $\lambda^{-1}$ |
|---|---|---|---|
| 0.5 | 0.015(3) | 0.007(6) | 0.007(6) |
| 0.6 | 0.053(8) | 0.026(9) | 0.026(9) |
| 0.7 | 0.122(5) | 0.061(8) | 0.061(8) |
| 0.8 | 0.229(3) | 0.112(1) | 0.112(0) |
| 0.9 | 0.340(8) | 0.170(7) | 0.169(9) |

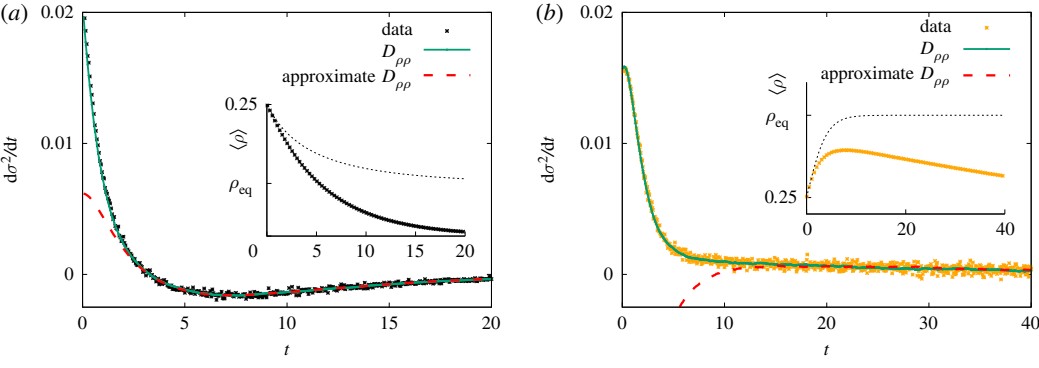

**Figure 8.** Change rate of $\sigma^2(t)$. Simulations with $N = 20$ agents ($10^6$ samples). (a) $\gamma/\alpha = 0.9$. Forward derivative data are consistent with predictions using equation (4.4) (solid line). The validity of the approximation $D_{\rho\rho} = -D_1/\langle\rho\rangle$ is restricted to non-Gaussian regime (red dashed line). Inset: $\langle\rho(t)\rangle$ quickly deviates from classical predictions of compartmental equation (dashed line). (b) $\gamma/\alpha = 0.5$. Non-Gaussian regimes takes a lot longer to start.

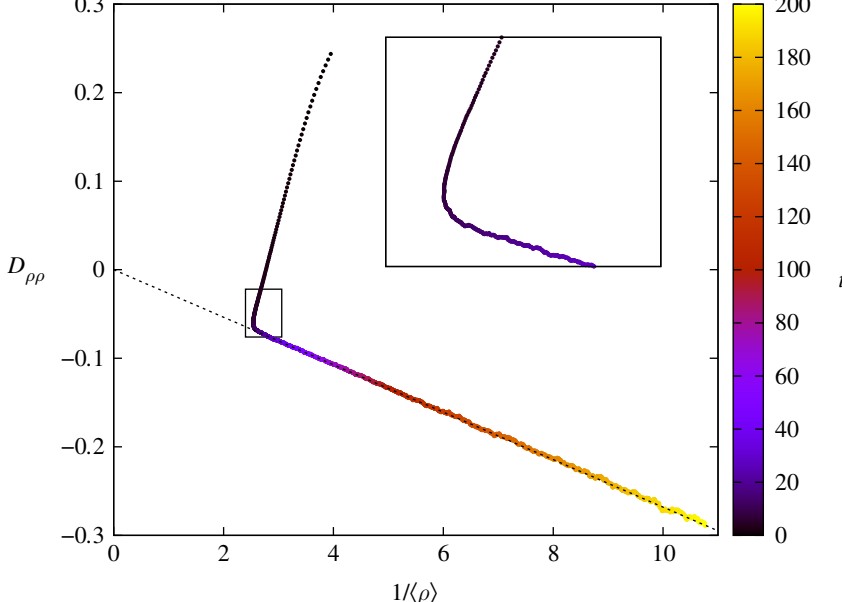

**Figure 9.** Evolution of $D_{\rho\rho}$ with $\langle\rho\rangle^{-1}$. Data are colour coded with time. The region inside the rectangle demarks the time interval corresponding to the transition between distinct fluctuation regimes. The linear relationship between $D_{\rho\rho}$ and $\langle\rho\rangle^{-1}$ dictates the system evolution, in the non-Gaussian regime. The dotted line depicts the corresponding line equation that crosses the origin.

By virtue of equation (4.3), we now exploit the relationship between $\Delta_3(t)$ and $D_{\rho\rho}(t)$ to propose an equation for the expected dynamics of non-Gaussian fluctuations in equation (3.5):

$$\frac{1}{2\alpha}\frac{\mathrm{d}\sigma^2}{\mathrm{d}t} = \langle\rho\rangle\sigma^2 + [\langle\rho\rangle - \rho_{\mathrm{eq}} + D_{\rho\rho}(t)]\langle\rho\rangle^2 + \frac{s(t)}{N}, \tag{4.4}$$

where $s(t) = [(2 - \langle\rho\rangle - \rho_{\mathrm{eq}})\langle\rho\rangle/2] - \sigma^2(t)$. Equation (4.4) agrees well with simulated data for the entire time interval considered (figure 8) However, the same agreement is not observed for the approximate formula $D_{\rho\rho}(t) = -D_1 e^{t/\xi}$ for the entire time interval. In fact, away from the non-Gaussian regime where the fit is accurate, most of the data fall off the proposed curve. Figure 9 explains the reason: $D_{\rho\rho}[\rho] = a\langle\rho\rangle^{-1} + b$ is a straight line that intercepts the origin only after the time interval enclosed by the rectangle. The width of the segment shrinks with increasing values of $N$. Prior to this interval, the curve $D_{\rho\rho}[\rho]$ slightly deviates from a straight line, with non-vanishing intercept $b$. Although an estimate of $D_{\rho\rho}[\rho]$ could be useful in this regime, a far more accurate calculation can be obtained by different means, as we show in the next section.

For practical purposes, one can either monitor how $D_{\rho\rho}^2(t)/\langle\rho(t)\rangle^2$ evolves along time or $D_{\rho\rho}$ as a function of $\langle\rho\rangle^{-1}$. Both methods capture the transition between fluctuation regimes. In the

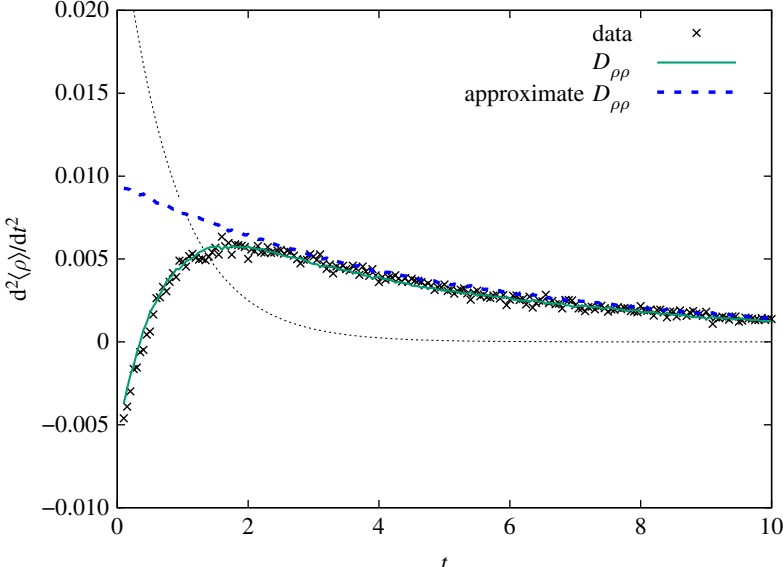

**Figure 10.** Second-order differential equation. Numerical simulations with $N = 20$ agents, $\gamma/\alpha = 0.9$, and $10^6$ samples. The exact formula in equation (4.5) agrees with simulated data for arbitrary $t$. The approximation $D_{\rho\rho} = -D_1/\langle\rho\rangle$ fails to replicate the data during initial times (thick dashed line). The dotted line represents the compartmental prediction $d^2\rho/dt^2 = \alpha^2 (\rho_{eq} - 2\rho)(\rho_{eq} - \rho)\rho$, obtained by taking the derivative of equation (2.1).

non-Gaussian regime, one should use equations (3.4) and (4.4). Approximations for $D_{\rho\rho}(t)$ should be used with care as indicated in figure 8.

For the sake of completeness, we derive the second-order differential equation for $\langle\rho(t)\rangle$. Taking the time derivative of equation (3.4) and using equation (4.4), one arrives at the desired expression

$$\frac{d^2\langle\rho\rangle}{dt^2} = \alpha\rho_{eq}\frac{d\langle\rho\rangle}{dt} - 2\alpha^2\langle\rho\rangle^2 D_{\rho\rho}(t) - \frac{2\alpha^2 s(t)}{N}. \tag{4.5}$$

Results show an excellent agreement with simulated data, regardless of fluctuation type (figure 10). Furthermore, one can employ the approximation $D_{\rho\rho} \approx -D_1/\langle\rho(t)\rangle$, with $D_1 \geq 0$ for fixed $N$ and epidemiological parameters as well. Under this assumption, agreement is observed only in the non-Gaussian regime, as expected. It is instructive to study equation (4.5) when $o(1/N)$ corrections are neglected

$$\frac{d^2\langle\rho\rangle}{dt^2} \approx \alpha\rho_{eq}\frac{d\langle\rho\rangle}{dt} + 2\alpha^2 D_1\langle\rho\rangle. \tag{4.6}$$

The characteristic equation provides a coarse estimate for

$$\xi_{est}^{-1} = -\frac{\alpha\rho_{eq}}{2}\left[1 - \left(1 + \frac{8D_1}{\rho_{eq}}\right)^{1/2}\right]. \tag{4.7}$$

This expression allows one to quickly grasp the dependence between $\xi_{est}$ and the parameter $D_1$. However, there are several issues with $\xi_{est}$. The most important one deals with the hypothesis that $o(1/N)$ terms contribute less than other terms in equation (4.5). In fact, they are similar in magnitude and should not be discarded. A far more reliable estimate can be obtained assuming $\sigma^2(t)$ can be written as a power series, i.e. $\sigma^2(t) \approx \sum_{m=1} \sigma_m^2 e^{-m(t/\xi)}$. Collecting only terms proportional to $e^{-t/\xi}$, one deduces $D_1$ in equation (4.7) should be replaced by $D_1 + (1/N)[(\sigma_1^2/\rho_1) - (2 - \rho_{eq})/2]$. For instance, numerical data suggest $\xi_{est} = 0.201$ ($N = 20$ and $\gamma/\alpha = 0.9$), with $D_1 = 0.065$, $\rho_1 = 0.25$, $\sigma_1 = 0.063$.

# 5. Gaussian fluctuations

For large population sizes $N \gg 1$, stochastic effects are well represented by Gaussian fluctuations and dominated by finite second moments. Noting that the skewness coefficient $\kappa_3 = (\Delta_3 - 3\langle\rho\rangle\sigma^2)/\sigma^3$ vanishes for Gaussian distributions, we conclude $\Delta_3^{gauss} \approx 3\langle\rho(t)\rangle\sigma^2(t)$. Indeed, figure 5 shows the ansatz is not too far-fetched since $\Delta_3(t) - \Delta_3^{gauss}(t) \sim o(\sigma^2/N)$ for ratios $\gamma/\alpha = 0.1$ and $0.5$ for $N = 50$.

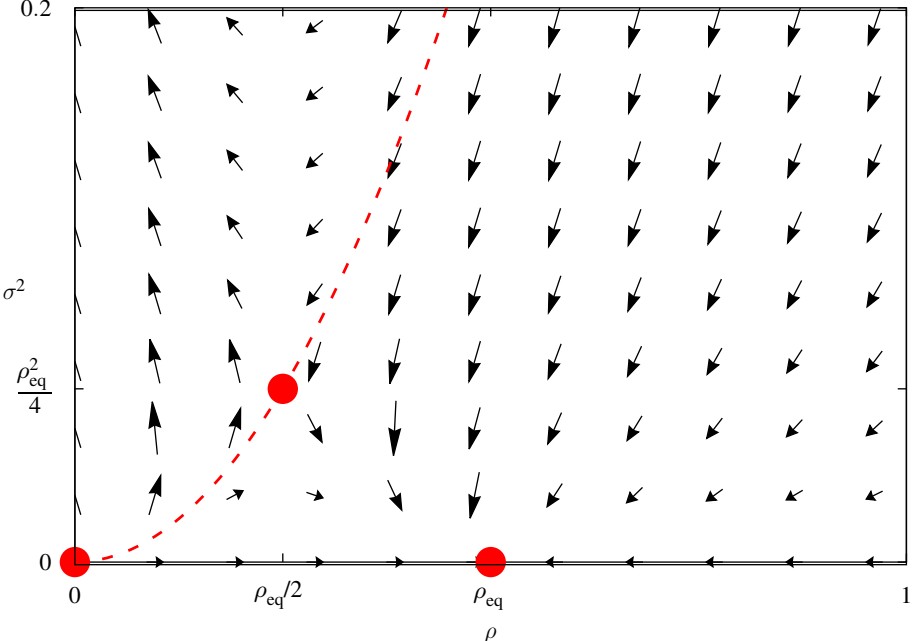

**Figure 11.** Direction field and critical points. The critical points (red circles) in the phase plane $(\rho, \sigma^2)$ are $(0, 0)$, $(\rho_{eq}, 0)$, and $(\rho_{eq}/2, \rho_{eq}^2/4)$. The first two critical points are equilibrium points for the usual compartmental equation, while the remaining one lies at the separatrix $\sigma^2 = \rho^2$ (dashed line). Above the separatrix, equations (5.1a) and (5.1b) fail to converge.

Ignoring $o(1/N)$ corrections in equation (3.5), the following differential equations are obtained:

$$\frac{1}{\alpha}\frac{\mathrm{d}}{\mathrm{d}t}\ln\langle\rho\rangle = \rho_{eq} - \langle\rho\rangle - \frac{\sigma^2}{\langle\rho\rangle} \tag{5.1a}$$

and

$$\frac{1}{2\alpha}\frac{\mathrm{d}}{\mathrm{d}t}\ln\sigma^2 = \rho_{eq} - 2\langle\rho\rangle. \tag{5.1b}$$

Both equations have been derived previously (see [26,27]). As long as $\sigma^2(0) > 0$, uncertainties play a role in the SIS epidemic model; $\sigma^2(0) = 0$ implies $\sigma^2(t) = 0$ and warrants the validity of equation (2.2). Thus, the instantaneous factor $\sigma^2(t)/\langle\rho(t)\rangle$ in equation (5.1a) improves compartmental predictions if $\sigma^2(t) \neq 0$. Figure 11 portrays the corresponding direction field.

Despite the insights provided by equations (5.1a) and (5.1b), some issues still remains. The most relevant one deals with the evaluation of $\sigma^2(0)$ from real epidemiological data. In essence, $\sigma^2(0)$ encapsulates the measurement of ignorance about the system at $t = 0$. In practice, one would rely on clever measurements—possibly, with bias—to estimate $\sigma^2(0)$. Alternatively, the issue can be avoided entirely by combining the system of differential equations for $\rho(t)$ and $\sigma^2(t)$ into a single differential equation

$$\frac{\mathrm{d}^2\langle\rho\rangle}{\mathrm{d}t^2} = 3\alpha(\rho_{eq} - 2\langle\rho\rangle)\left[\frac{\mathrm{d}\langle\rho\rangle}{\mathrm{d}t} - \frac{2\alpha}{3}\langle\rho\rangle(\rho_{eq} - \langle\rho\rangle)\right]. \tag{5.2}$$

Recalling that setting $\sigma^2(t) = 0$ is equivalent to using compartmental equations, one can borrow inspiration from projective transformations and rational functions to search for solutions of equation (5.2) starting from equation (2.1). More specifically, assume

$$\frac{\langle\rho(t)\rangle}{\rho_{eq}} = \frac{\sum_{k=0}^{m} a_k x^k(t)}{\sum_{\ell=0}^{m} b_\ell x^\ell(t)}, \tag{5.3}$$

with $x(t) = e^{-\alpha\rho_{eq}t}$. The coefficients $a_k$ and $b_k$ are obtained for fixed integer $m$, in consonance with the three critical points discussed earlier, and with the solution of compartmental equations (case $\sigma^2 = 0$).

A suitable candidate is

$$\frac{\langle\rho(t)\rangle}{\rho_{eq}} = \frac{a_0 + a_1 e^{-\alpha\rho_{eq}t}}{b_0 + b_1 e^{-\alpha\rho_{eq}t} + b_2 e^{-2\alpha\rho_{eq}t}}. \tag{5.4}$$

Plugging the expression above in equation (5.2) and solving the coefficients, one obtains two solutions in addition to the trivial solutions. The analytical solution that encircles all values $(\rho, \sigma^2)$ below the separatrix, for $\alpha > \gamma$, reads

$$\frac{\langle\rho(t)\rangle_1}{\rho_{eq}} = \frac{1 + a_1 e^{-\alpha\rho_{eq}t}}{1 + 2a_1 e^{-\alpha\rho_{eq}t} + b_2 e^{-2\alpha\rho_{eq}t}}, \tag{5.5}$$

with constants $a_1$ and $b_2$ determined by initial conditions $\rho(0)$ and $(d\rho/dt)_{t=0}$. Note that $\sigma^2(t)$ can be computed from equation (5.1a), $\sigma_1^2(t) = \langle\rho(t)\rangle_1^2(a_1^2 - b_2)e^{-2\alpha\rho_{eq}}$. The constraint $\sigma_1^2(t) \geq 0$ implies $a_1^2 - b_2 \geq 0$, while the solution of the compartmental equation is obtained by setting $b_2 = a_1^2$.

The remaining solution is

$$\frac{\langle\rho(t)\rangle_2}{\rho_{eq}} = \frac{1}{2 + b_1 e^{-\alpha\rho_{eq}t}}, \tag{5.6}$$

with $b_1 = [\rho_{eq}/\rho(0)] - 2$. It corresponds to the case $\sigma_2^2(t) = \rho_2^2(t)$ and includes the third critical point $(\rho_{eq}/2, \rho_{eq}^2/4)$, along the separatrix. The role of the separatrix can be understood in terms of the signal-to-noise ratio $s(t) = \langle\rho(t)\rangle^2/\sigma^2(t)$. Below the separatrix, $s(t) > 1$ and the average $\langle\rho(t)\rangle$ becomes more relevant than $\sigma^2(t)$, leading to the equilibrium density $\rho_{eq}$. At the separatrix, $s(t) = 1$ and it indicates that both signal and noise are present in equal measures. Indeed, at the critical point $(\rho_{eq}/2, \rho_{eq}^2/4)$ one would expect $\langle\rho(t \gg 1)\rangle$ to fluctuate around $\rho_{eq}/2$, confined between the other critical points. For large $\rho_{eq}$, it also means large deviations. By contrast, noise becomes predominant for $s(t) < 1$, leading to non-biological dynamics as illustrated in figure 11. Perhaps one can argue $s(t) < 1$ implies some of the samples used to calculate $\langle\rho\rangle$ acquire negative values. In this case, the Gaussian description becomes inadequate to portray the biological system. To reinforce this conclusion, one can consider the limiting case with vanishing $\langle\rho(t)\rangle$ but finite $\sigma^2(t)$: equation (5.1b) is approximated by $d\sigma^2/dt \approx 2\rho_{eq}\sigma^2$ so that $\sigma^2(t) = \sigma^2(0)e^{2\rho_{eq}t}$. Therefore, the negative parcel in equation (5.1a) grows exponentially along time, producing negative solutions.

The main point of equation (5.2) relies on its compatibility with day-to-day epidemiological data, usually built upon the number of infected patients within a fixed time window. Furthermore, mathematical properties of equations (5.1–5.5) lie well beyond the scope of this paper and merit a proper discussion elsewhere.

# 6. Conclusion

We have investigated the effects of uncertainties in the SIS epidemic model, finding new differential equations for the average density of infected agents, $\rho(t)$, and its corresponding variance, $\sigma^2(t)$. Our findings reconcile the simplicity of canonical compartmental equations with the accuracy of agent-based simulations, creating suitable tools for practitioners of epidemiology and related fields. At the core of this research, we have demonstrated that uncertainty cannot be neglected in the SIS epidemic model whenever the discreteness of the population is important, even when the population comprises statistically equivalent agents. Uncertainties are inherent aspects of stochastic spreading processes, and their time evolutions are key elements to describe how the number of infected agents vary along time. Concerning their nature specifically, numerical simulations in fully connected networks reveal that uncertainties can be organized into two broad classes, namely, Gaussian and non-Gaussian fluctuations. Gaussian fluctuations, also known as symmetric fluctuations, dominate the spreading process whenever $\langle\rho(t)\rangle$ and $\sigma^2(t)$ are sufficient to describe the outbreak. This scenario implies the skewness coefficient vanishes for large $N$, producing a simplified system of differential equations for $\langle\rho(t)\rangle$ and $\sigma^2(t)$. Alternatively, the differential equations can be combined into a second-order differential equation for $\langle\rho(t)\rangle$, avoiding problems due to poor estimates of initial values of $\sigma^2(t)$ from raw data. Non-Gaussian fluctuations are far more complex to assess, as they emerge as a consequence of large recovery rates in small population. More specifically, the stochastic process tends to perceive the influence of the absorbing state $\rho = 0$, creating asymmetric fluctuations. As a result, the skewness coefficient does not converge to a simple mathematical form. Instead, differential equations for $\langle\rho(t)\rangle$ and $\sigma^2(t)$ are written using the normalized autocorrelation function $D_{\rho\rho}(t)$. This function is relevant for the spreading process because it can be interpreted as the likelihood of adequate contact between a given infected agent with susceptible

ones, for vanishing variance and large population sizes. For non-Gaussian fluctuations, our numerical simulations show that $D_{\rho\rho}(t)$ is proportional $\langle \rho(t) \rangle$. Therefore, the spreading process reduces, again, to a closed system of differential equations for $\langle \rho(t) \rangle$ and $\sigma^2(t)$ (see equation (4.4)). Finally, we stress that this research evaluates the impact of uncertainties only for homogeneous populations, i.e. connections between agents are described according to the complete graph. An intriguing question is left open concerning the role of uncertainties in disease spreading processes in other networks.

Numerical simulations are performed using the direct Monte Carlo method [40]. It shares the same origins as the Gillespie algorithm [41], differing only on time step selection. In the Gillespie algorithm, the time step is a random variable distributed according to an exponential pdf; and the system always suffers a single modification once the time step is selected, given by the off-diagonal elements of the transition matrix. In the direct Monte Carlo method, the time step is fixed and equal in value to the average time step of the Gillespie algorithm, both consistent with the Poisson hypothesis. Furthermore, after a single time step has elapsed the system has a chance to remain in the same configuration (diagonal elements of the transition matrix), in addition to the off-diagonal transitions. Direct Monte Carlo simulations tend to be slower than Gillespie but allow for a simple evaluation of statistics at discrete time, including their derivatives and autocorrelation functions, without additional processing algorithms or interpolation methods.

Ethics. This work does not present research with ethical considerations.

Data accessibility. Data and numerical codes are deposited at https://doi.org/10.17605/OSF.IO/WFCEP.

Authors' contributions. A.S.M., G.C.C. and G.M.N. designed the research; G.M.N. performed the research and wrote the paper; G.C.C. and A.S.M. edited the paper. All authors reviewed the manuscript.

Competing interests. The authors declare no competing interest.

Funding. This work is supported by Capes 88887.136416/2017-00 and CNPq 307948/2014-5.

Acknowledgements. We thank N.D. Gomes for her contributions during the earlier stages of the study. We are grateful to G. Contesini and F. Meloni for comments during the manuscript preparation and subsequent discussions.

# Appendix A. Equations of motion

Let $\hat{\eta} = \sum_{k=0}^{N-1} \hat{n}_k$. The equations of motion for $\langle \rho(t) \rangle$ and $\langle \rho^2(t) \rangle$ are, respectively,

$$
\begin{aligned}
\frac{\mathrm{d}\langle\rho\rangle}{\mathrm{d}t} &= -\frac{1}{N}\sum_{v=0}^{2^N-1}\langle\mu|\hat{\eta}\hat{H}|v\rangle P_v(t) \\
&= -\frac{\alpha}{N^2}\sum_{\mu,v=0}^{2^N-1} P_v(t)\left[\langle\mu|\hat{\eta}(N-\hat{\eta})\hat{\eta}|v\rangle - \sum_{k=0}^{N-1}\langle\mu|\hat{\eta}(\hat{\sigma}_k^+)\hat{\eta}|v\rangle\right] + \\
&\quad -\frac{\gamma}{N}\sum_{\mu,v=0}^{2^N-1} P_v(t)\left[\langle\mu|\hat{\eta}^2|v\rangle - \sum_{k=0}^{N}\langle\mu|\hat{\eta}(\hat{\sigma}_k^-)|v\rangle\right] \\
&= -\frac{\alpha}{N^2}\sum_{v=0}^{2^N-1} P_v(t)[\eta_v^2(N-\eta_v) - (\eta_v+1)(N-\eta_v)\eta_v] + \\
&\quad -\frac{\gamma}{N}\sum_{v=0}^{2^N-1} P_v(t)[\eta_v^2 - (\eta_v-1)\eta_v] \\
&= +\alpha\langle\rho(t)[1-\rho(t)]\rangle - \gamma\langle\rho(t)\rangle
\end{aligned}
\tag{A 1a}
$$

and

$$
\begin{aligned}
\frac{\mathrm{d}\langle\rho^2\rangle}{\mathrm{d}t} &= -\frac{1}{N^2}\sum_{\mu,v=0}^{2^N-1}\langle\mu|\hat{\eta}^2\hat{H}|v\rangle \\
&= -\frac{\alpha}{N^3}\sum_{v=0}^{2^N-1} P_v(t)[(N-\eta_v)\eta_v^3 - (N-\eta_v)(\eta_v+1)^2\eta_v] + \\
&\quad -\frac{\gamma}{N^2}\sum_{v=0}^{2^N-1} P_v(t)[\eta_v^3 - (\eta_v-1)^2\eta_v] \\
&= +2\alpha\langle\rho^2(1-\rho)\rangle - 2\gamma\langle\rho^2\rangle + \frac{\alpha}{N}\langle\rho(1-\rho)\rangle + \frac{\gamma}{N}\langle\rho\rangle.
\end{aligned}
\tag{A 1b}
$$

Since $\sigma^2(t) = \langle\rho^2(t)\rangle - \langle\rho(t)\rangle^2$, the differential equation for $\sigma^2(t)$ reads

$$\frac{1}{2}\frac{\mathrm{d}\sigma^2}{\mathrm{d}t} = \frac{1}{2}\frac{\mathrm{d}\langle\rho^2\rangle}{\mathrm{d}t} - \langle\rho\rangle\frac{\mathrm{d}\langle\rho\rangle}{\mathrm{d}t}$$

$$= \alpha[\rho_{\mathrm{eq}} + \langle\rho\rangle]\sigma^2 - \alpha\Delta_3(t) + \frac{\alpha}{2N}\langle\rho(1-\rho)\rangle + \frac{\gamma}{2N}\langle\rho\rangle, \tag{A 2}$$

where $\Delta_3(t) = \langle\rho^3\rangle - \langle\rho\rangle^3$.

# Appendix B. Evaluation of $C_{\rho\rho}(t)$

According to the definition in equation (4.1), we only need to calculate $\langle\rho(t+\delta)\rho(t)\rangle$. Up to $o(\delta t^2)$, following equation (4.2),

$$\langle\rho(t+\delta)\rho(t)\rangle = \langle\rho^2\rangle - \frac{\delta t}{N^2}\sum_{\mu,\nu=0}^{2^N-1}P_\nu(t)\eta_\nu\eta_\mu\langle\mu|\hat{H}|\nu\rangle + o(\delta t^2)$$

$$= \langle\rho^2\rangle - \frac{\alpha\delta t}{N^3}\sum_{\nu=0}^{2^N-1}P_\nu(t)[\eta_\nu^3(N-\eta_\nu) - \eta_\nu^2(N-\eta_\nu)(\eta_\nu+1)]$$

$$- \frac{\gamma\delta t}{N^2}\sum_{\nu=0}^{2^N-1}P_\nu(t)[\eta_\nu^3 - \eta_\nu^2(\eta_\nu-1)] + o(\delta t^2)$$

$$= \langle\rho^2\rangle + \alpha\delta t[\rho_{\mathrm{eq}}\langle\rho^2\rangle - \langle\rho^3\rangle] + o(\delta t^2). \tag{B 1}$$

Hence, $C_{\rho\rho}(t) = \sigma^2(t) + \alpha\delta t\ [\rho_{\mathrm{eq}}\langle\rho^2\rangle - \langle\rho^3\rangle] + o(\delta t^2)$.

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
