## [Reviewer comments · Royal Society Open Science]

Review History

RSOS-182156.R0 (Original submission)

Review form: Reviewer 1

Is the manuscript scientifically sound in its present form?

No

Are the interpretations and conclusions justified by the results?

No

Is the language acceptable?

Yes

Is it clear how to access all supporting data?

No

Do you have any ethical concerns with this paper?

No

Have you any concerns about statistical analyses in this paper?

No

Recommendation?

Major revision is needed (please make suggestions in comments)

Comments to the Author(s)

See attached PDF (Appendix A).

Review form: Reviewer 2

Is the manuscript scientifically sound in its present form?

Yes

Are the interpretations and conclusions justified by the results?

Yes

Is the language acceptable?

Yes

Is it clear how to access all supporting data?

Not Applicable

Do you have any ethical concerns with this paper?

No

Have you any concerns about statistical analyses in this paper?

I do not feel qualified to assess the statistics

Recommendation?

Major revision is needed (please make suggestions in comments)

Comments to the Author(s)

Review of "RSOS-182156"

The paper proposes a mean-field models for the SIS epidemic on a fully connected network by taking into consideration the higher order terms which emerge during the averaging out the mean number of infected nodes at time "t" from the exact stochastic model. This higher order term has an equation of its own which then depends on even higher order terms. The authors propose to break this dependency by approximating the third moment of number of infectious individuals at time t. They make some approximations based on the assumptions that the fluctuations are gaussian. The paper has some interesting elements, but overall I have several major issues:

1. First of all, I invite the authors to have a look at the work done in the paper titled "New Moment Closures Based on A Priori Distributions with Applications to Epidemic Dynamics" Bull Math Biol (2012) 74:1501-1515 DOI 10.1007/s11538-012-9723-3. In this paper the authors consider the SIS model on a fully connected network and derive ODEs for the first and second moments and propose the closure of the third moment in terms of the first and second. This is based on the assumption that $p_k(t)$ (the probability of observing k infected individuals at time "t") is binomially distributed. They end up with a system of two ODEs. This is very similar to what is done in this new paper. They also show numerically that the difference between the exact system and the closed system seem to scale like $1/N^2$, which is an improvement over $1/N$ for some previously use closures. The normal distribution is also proposed as a potential candidate rather

than the binomial. So I am not too sure that I see where the novelty of the paper lies and how it builds/adds/complements the state-of-the-art.

2. There is little merit in simulating epidemics on graphs of size 50, since one has exact solutions by solving 51 linear ODEs, this can be done easily even for network where the number of nodes scale like $O(1000)$. This can be also used to compare the moments of the true model and that of the approximation.

3. Since the whole analysis focuses on the fully connected network, there is not point in introducing overly complicated notation and talking about state space of size $2^{\{N\}}$. In this case, the exact stochastic model is given by the forward Kolmogorov equations with $N+1$ states, and this is very well known. Even more well-known is Eq. (7) and this does no need to be derived, see of example Epidemic Modelling by Daley and Gani (done for SIR but is identical for SIS) or Mathematics of epidemics on networks by Kiss, Simon and Joel. Eq. (7) can simply be stated and referenced accordingly, or the paper cited at point (1).

4. In the text below figure 1, the authors use reference [23] which seems to be their own, but again these facts are well known for some time, see also the second reference that I provided above or others. But again, I believe since the whole paper is based on the fully connected network, the very technical notation is not needed.

5. Above equation (2) could the authors provide a reference for the extension of the α to $\alpha * \text{second}/\text{first moment}$? I am not sure this is totally correct.

6. Just above Section II (Complete Graph), the authors talk about simulation results but there is no mention how these are done, do they use the Gillespie algorithm or how are they done. I really would like to see comparisons between the average number of infected nodes taken from different model, rather than some transformed quantity.

7. The authors go on about and promote this new form of the equation but then they do not fit the model to any real-world epidemic, so why cast then the equation in an unfamiliar form if its usefulness is not shown.

8. In figure 2, has the Δ_3 been taken from simulation? Again, I would plot the average number of infected nodes and not only the approximations.

9. I could not get Eq 9b, is there a σ^2 missing on either the left- or right-hand side?

10. The Gaussian approximation only seem to work for some parameter values. Can the authors map this out, otherwise the model is not very useful as its range of operation is not known.

11. Effectively one needs to use two different models based on what values α and γ take.

12. The stochastic version of this model is analysed in detail in the seminal book by Nasell (Extinction and quasistationarity in the stochastic Logistic SIS model).

13. The authors motivate their work by taking about populations of small size yet in the derivation $N \gg 1$ everywhere. I am confused.

Overall, I believe the paper seems a bit confused and unclear about the findings and usefulness of the new model and the authors are not aware of the relevant literature. To be regarded as a good model, I would like to see extensive numerical simulations and tests by comparing the average number of infected nodes between simulations and the mean field model for a large range of α and γ and even different population size. More importantly, I would like to see if this new closure produces a better model when comparing to other closures that exist in the literature.

Decision letter (RSOS-182156.R0)

10-Jun-2019

Dear Dr Nakamura:

Manuscript ID RSOS-182156 entitled "Improved SIS epidemic equations based on uncertainties and autocorrelation functions" which you submitted to Royal Society Open Science, has been reviewed. The comments from reviewers are included at the bottom of this letter.

In view of the criticisms of the reviewers, the manuscript has been rejected in its current form. However, a new manuscript may be submitted which takes into consideration these comments.

Please note that resubmitting your manuscript does not guarantee eventual acceptance, and that your resubmission will be subject to peer review before a decision is made.

Your resubmitted manuscript should be submitted by 08-Dec-2019. If you are unable to submit by this date please contact the Editorial Office.

on behalf of Dr Anna Marciniak-Czochra (Associate Editor) and Mark Chaplain (Subject Editor)
openscience@royalsociety.org

Reviewers' Comments to Author:
Reviewer: 1

Comments to the Author(s)
See attached PDF

Reviewer: 2

Comments to the Author(s)
Review of "RSOS-182156"

The paper proposes a mean-field models for the SIS epidemic on a fully connected network by taking into consideration the higher order terms which emerge during the averaging out the mean number of infected nodes at time "t" from the exact stochastic model. This higher order term has an equation of its own which then depends on even higher order terms. The authors

propose to break this dependency by approximating the third moment of number of infectious individuals at time t . They make some approximations based on the assumptions that the fluctuations are gaussian. The paper has some interesting elements, but overall I have several major issues:

1. First of all, I invite the authors to have a look at the work done in the paper titled "New Moment Closures Based on A Priori Distributions with Applications to Epidemic Dynamics" Bull Math Biol (2012) 74:1501–1515 DOI 10.1007/s11538-012-9723-3. In this paper the authors consider the SIS model on a fully connected network and derive ODEs for the first and second moments and propose the closure of the third moment in terms of the first and second. This is based on the assumption that $p_k(t)$ (the probability of observing k infected individuals at time " t ") is binomially distributed. They end up with a system of two ODEs. This is very similar to what is done in this new paper. They also show numerically that the difference between the exact system and the closed system seem to scale like $1/N^2$, which is an improvement over $1/N$ for some previously use closures. The normal distribution is also proposed as a potential candidate rather than the binomial. So I am not too sure that I see where the novelty of the paper lies and how it builds/adds/complements the state-of-the-art.
2. There is little merit in simulating epidemics on graphs of size 50, since one has exact solutions by solving 51 linear ODEs, this can be done easily even for network where the number of nodes scale like $O(1000)$. This can be also used to compare the moments of the true model and that of the approximation.
3. Since the whole analysis focuses on the fully connected network, there is not point in introducing overly complicated notation and talking about state space of size $2^{\{N\}}$. In this case, the exact stochastic model is given by the forward Kolmogorov equations with $N+1$ states, and this is very well known. Even more well-known is Eq. (7) and this does no need to be derived, see of example Epidemic Modelling by Daley and Gani (done for SIR but is identical for SIS) or Mathematics of epidemics on networks by Kiss, Simon and Joel. Eq. (7) can simply be stated and referenced accordingly, or the paper cited at point (1).
4. In the text below figure 1, the authors use reference [23] which seems to be their own, but again these facts are well known for some time, see also the second reference that I provided above or others. But again, I believe since the whole paper is based on the fully connected network, the very technical notation is not needed.
5. Above equation (2) could the authors provide a reference for the extension of the α to $\alpha * \text{second}/\text{first moment}$? I am not sure this is totally correct.
6. Just above Section II (Complete Graph), the authors talk about simulation results but there is no mention how these are done, do they use the Gillespie algorithm or how are they done. I really would like to see comparisons between the average number of infected nodes taken from different model, rather than some transformed quantity.
7. The authors go on about and promote this new form of the equation but then they do not fit the model to any real-world epidemic, so why cast then the equation in an unfamiliar form if its usefulness is not shown.
8. In figure 2, has the Δ_3 been taken from simulation? Again, I would plot the average number of infected nodes and not only the approximations.
9. I could not get Eq 9b, is there a σ^2 missing on either the left- or right-hand side?
10. The Gaussian approximation only seem to work for some parameter values. Can the authors map this out, otherwise the model is not very useful as its range of operation is not known.

11. Effectively one needs to use two different models based on what values α and γ take.

12. The stochastic version of this model is analysed in detail in the seminal book by Nasell (Extinction and quasistationarity in the stochastic Logistic SIS model).

13. The authors motivate their work by talking about populations of small size yet in the derivation $N \gg 1$ everywhere. I am confused.

Overall, I believe the paper seems a bit confused and unclear about the findings and usefulness of the new model and the authors are not aware of the relevant literature. To be regarded as a good model, I would like to see extensive numerical simulations and tests by comparing the average number of infected nodes between simulations and the mean field model for a large range of α and γ and even different population size. More importantly, I would like to see if this new closure produces a better model when comparing to other closures that exist in the literature.

Author's Response to Decision Letter for (RSOS-182156.R0)

See Appendix B.

RSOS-191504.R0

Review form: Reviewer 1

Is the manuscript scientifically sound in its present form?

Yes

Are the interpretations and conclusions justified by the results?

Yes

Is the language acceptable?

Yes

Do you have any ethical concerns with this paper?

No

Have you any concerns about statistical analyses in this paper?

No

Recommendation?

Accept as is

Comments to the Author(s)

In the revised version, the authors describe more clearly what they have done.

I am still not convinced by their answer to my comment that ensemble averages are not observable quantities. In the response, they acknowledge the issue but state that "from a methodological perspective, however, the underlying assumption of an ensemble (either from truly independent samples, or weakly interacting populations) is necessary to establish the nature of the uncertainties in the system." Ok, but they are not quite answering.

Anyway, I am not convinced of the usefulness of the authors' approach to the problem, but others may think it otherwise. In that perspective, the computations presented in the manuscript may provide a useful contribution.

I have only one small observation to the text:

- at p.11 l.195-196 the authors write "The effect can be found in small populations but it is enhanced in small populations:" There must be a typo.

Review form: Reviewer 2

Is the manuscript scientifically sound in its present form?

Yes

Are the interpretations and conclusions justified by the results?

Yes

Is the language acceptable?

Yes

Do you have any ethical concerns with this paper?

No

Have you any concerns about statistical analyses in this paper?

No

Recommendation?

Accept with minor revision (please list in comments)

Comments to the Author(s)

The authors acknowledged my comments and made some changes as a result.

Was disappointed that at least in two cases even though they agreed with my comments they did not implement them fully, see for example:

Environment - two-monthly review cycle

Next major advance would be done prior to the ERA visit.

To be reviewed by PA RSG.

Regular monthly RSG for 1 hour, progress review every two months

RSG will be particularly looking at the Outputs and Environment.

The first meeting likely to be scheduled mid-late September, which will give us chance to consider where we think we are, before term starts and before we receive the feedback from the University.

and

Environment - two-monthly review cycle

Next major advance would be done prior to the ERA visit.

To be reviewed by PA RSG.

Regular monthly RSG for 1 hour, progress review every two months

RSG will be particularly looking at the Outputs and Environment.

The first meeting likely to be scheduled mid-late September, which will give us chance to consider where we think we are, before term starts and before we receive the feedback from the University.

and

they still did not produce simulations where the agreement between the expected number of infected individuals is shown.

Decision letter (RSOS-191504.R0)

09-Dec-2019

Dear Dr Nakamura,

The Subject Editor assigned to your paper ("Improved SIS epidemic equations based on uncertainties and autocorrelation functions") has now received comments from reviewers. We would like you to revise your paper in accordance with the referee and Associate Editor suggestions which can be found below (not including confidential reports to the Editor). Please note this decision does not guarantee eventual acceptance.

Please submit a copy of your revised paper before 01-Jan-2020. Please note that the revision deadline will expire at 00.00am on this date. If we do not hear from you within this time then it will be assumed that the paper has been withdrawn. In exceptional circumstances, extensions may be possible if agreed with the Editorial Office in advance. We do not allow multiple rounds of revision so we urge you to make every effort to fully address all of the comments at this stage. If deemed necessary by the Editors, your manuscript will be sent back to one or more of the original reviewers for assessment. If the original reviewers are not available we may invite new reviewers.

When submitting your revised manuscript, you must respond to the comments made by the referees and upload a file "Response to Referees" in "Section 6 - File Upload". Please use this to document how you have responded to each of the comments, and the adjustments you have made. In order to expedite the processing of the revised manuscript, please be as specific as possible in your response.

- Ethics statement

- Data accessibility

<http://datadryad.org/submit?journalID=RSOS&manu=RSOS-191504>

- Competing interests

- Authors' contributions

- Acknowledgements

- Funding statement

Kind regards,
Anita Kristiansen
Editorial Coordinator
Royal Society Open Science
openscience@royalsociety.org

on behalf of Mark Chaplain (Subject Editor)
openscience@royalsociety.org

Associate Editor Comments to Author:

Comments to the Author:

Thank you kindly for resubmitting your manuscript to Royal Society Open Science. Following peer review, we have received two referee reports on your manuscript.

Although both referees note that there have been improvements and changes made, both referees agree that some of their concerns have not been properly addressed, or addressed to their fullest potential. Particularly, Referee #1 states that an answer to their concern has not quite been provided, and Referee #2 states that although the authors agreed with the concern, it was not implemented.

Please ensure you take another look at the referee's comments and ensure that you sufficiently

provide information on what changes have and have not been made, and why you have or have not implemented suggested changes from the referees.

Reviewer comments to Author:

Reviewer: 2

Comments to the Author(s)

The authors acknowledged my comments and made some changes as a result.

Was disappointed that at least in two cases even though they agreed with my comments they did not implement them fully, see for example:

Environment - two-monthly review cycle

Next major advance would be done prior to the ERA visit.

To be reviewed by PA RSG.

Regular monthly RSG for 1 hour, progress review every two months

RSG will be particularly looking at the Outputs and Environment.

The first meeting likely to be scheduled mid-late September, which will give us chance to consider where we think we are, before term starts and before we receive the feedback from the University.

and

Environment - two-monthly review cycle

Next major advance would be done prior to the ERA visit.

To be reviewed by PA RSG.

Regular monthly RSG for 1 hour, progress review every two months

RSG will be particularly looking at the Outputs and Environment.

The first meeting likely to be scheduled mid-late September, which will give us chance to consider where we think we are, before term starts and before we receive the feedback from the University.

and

they still did not produce simulations where the agreement between the expected number of infected individuals is shown.

Reviewer: 1

Comments to the Author(s)

In the revised version, the authors describe more clearly what they have done.

I am still not convinced by their answer to my comment that ensemble averages are not observable quantities. In the response, they acknowledge the issue but state that "from a methodological perspective, however, the underlying assumption of an ensemble (either from truly independent samples, or weakly interacting populations) is necessary to establish the nature of the uncertainties in the system." Ok, but they are not quite answering.

Anyway, I am not convinced of the usefulness of the authors' approach to the problem, but others may think it otherwise. In that perspective, the computations presented in the manuscript may provide a useful contribution.

I have only one small observation to the text:

- at p.11 l.195-196 the authors write "The effect can be found in small populations but it is enhanced in small populations." There must be a typo.

Author's Response to Decision Letter for (RSOS-191504.R0)

See Appendix C.

RSOS-191504.R1 (Revision)

Review form: Reviewer 1

Is the manuscript scientifically sound in its present form?

Yes

Are the interpretations and conclusions justified by the results?

No

Is the language acceptable?

Yes

Do you have any ethical concerns with this paper?

No

Have you any concerns about statistical analyses in this paper?

No

Recommendation?

Accept as is

Comments to the Author(s)

As I wrote in my review of the previous version, I am not convinced of the usefulness of the authors' approach to the problem, but others may think it otherwise. In their response to my comments, the authors have added a sentence, that initially recognizes my doubts, and ends explaining the advantages provided by their method. Again, I do not follow them, but they made their point clearer.

Review form: Reviewer 2

Is the manuscript scientifically sound in its present form?

Yes

Are the interpretations and conclusions justified by the results?

Yes

Is the language acceptable?

Yes

Do you have any ethical concerns with this paper?

No

Have you any concerns about statistical analyses in this paper?

No

Recommendation?

Accept as is

Comments to the Author(s)

I think we reached a point where we have different views and opinions, probably each party has a point and what they say is correct in a certain setting but not at the same time. I still think that

the authors use a cannon to blow an ant, to use the author's words. But I am satisfied that the paper goes ahead and is published.

Decision letter (RSOS-191504.R1)

27-Jan-2020

Dear Dr Nakamura,

It is a pleasure to accept your manuscript entitled "Improved SIS epidemic equations based on uncertainties and autocorrelation functions" in its current form for publication in Royal Society Open Science. The comments of the reviewer(s) who reviewed your manuscript are included at the foot of this letter.

on behalf of Mr Andrew Dunn (Associate Editor) and Mark Chaplain (Subject Editor)
openscience@royalsociety.org

Reviewer comments to Author:
Reviewer: 2

Comments to the Author(s)
I think we reached a point where we have different views and opinions, probably each party has a point and what they say is correct in a certain setting but not at the same time. I still think that the authors use a cannon to blow an ant, to use the author's words. But I am satisfied that the paper goes ahead and is published.

Reviewer: 1

Comments to the Author(s)
As I wrote in my review of the previous version, I am not convinced of the usefulness of the authors' approach to the problem, but others may think it otherwise.

In their response to my comments, the authors have added a sentence, that initially recognizes my doubts, and ends explaining the advantages provided by their method. Again, I do not follow them, but they made their point clearer.

Appendix A

The manuscript “Improved SIS epidemic equations based on uncertainties and autocorrelation functions” by Nakamura *et al.* analyses a stochastic SIS model with homogeneous mixing, arriving at some approximate equations for the first two moments of the infected fraction.

I see two main weaknesses in the manuscript:

- The authors seem not to be very familiar with the literature on the topic, as can be seen by the lack of references to some of the main papers in which the model was studied, such as [KL89], [N96] and especially [N01]. Moreover, the use of correlation equations has been around for at least 25 years in the eco-epidemiological literature; for instance, equation (7) is (2) in [K00] (though in a different notation) and is at page 97 of the textbook [ME08]; the Gaussian approximation (9a)-(9b) is presented in Appendix A of [K00] while [N03a] and [N03b] are devoted to a systematic analysis of moment closure methods for the SIS stochastic model.
- In several points in the manuscript (e.g. at page 4) the authors discuss how the equations for the moments can be used for extracting epidemiological parameters from data, as if one could have observations of $\langle \rho(t) \rangle$, or of the variance σ^2 . However $\langle \rho(t) \rangle$ is an ensemble average, and could be estimated only if we had a large number of independent realizations of the epidemic process, as in the simulations presented in Fig. 2. In reality, one has some observations over a single realization of the epidemic process (assuming that the model is correct), and inference must be performed from this kind of data. It is not clear how equations (7), (9) or (17) can contribute to this. Probably it is for this reason that moment equations (although well known for a long time) have rarely been considered for processes with homogeneous mixing, but rather used to approximate spatial epidemics, or so-called metapopulation models, where one assumes there is a large number of populations with weak interactions (see for instance [BP97]).

I add some observations on specific points of the manuscript

1. The authors present the model as a continuous time Markov chain with 2^N states, corresponding to the N individuals. This may be necessary if contacts occur along some specific graphs; however, the authors analyse only the complete graph, in which individuals are indistinguishable, so that the process can be described on the state space $\{0, 1, \dots, N\}$, the number of infected individuals. This simplifies dramatically computations and notation (by the way, the authors could have made an effort to avoid the jargon of statistical physicists, and, when preparing a manuscript that should be widely read, to explain the notation used in that community).
2. The analysis of system (9) is incomplete. The authors are able to compute a class of explicit solutions (I am very surprised that the authors are able to find explicit solutions of a nonlinear second-order differential equation; honestly, I did not check that indeed (11) is a solution, suspecting that it must come

from some general approach), but they miss the fact that these are not all the solutions. Indeed, a simple analysis of (9) (shown in my drawing below) shows that there is a saddle point at $E^* = (\frac{\rho_{eq}}{2}, \frac{\rho_{eq}^2}{4})$. The stable manifold of E^* works as a separatrix: if initial conditions are below it, solutions asymptotically tend to $(\rho_{eq}, 0)$ and presumably are represented by (11); with initial conditions above the separatrix, $\langle \rho(t) \rangle$ becomes negative in finite time, so that the solutions lose biological realism. This fact was noted by [K00], who therefore suggested the use of multiplicative moments, based on lognormal approximation.

3. The authors discuss in Section 3 Gaussian fluctuations, and in Section 4 the use of autocorrelation functions, but they are very vague about the use of either, though they say that Gaussian fluctuations should be inappropriate when $\gamma/\alpha \approx 1$ and when population is small. This probably corresponds to the parameter regions with qualitatively different behaviours much more clearly identified in [N01].
4. Several arguments in Section 4 are hard to follow:
 - (a) 'Hence $D_{\rho\rho}(t)$ can be interpreted as an alternative metric to describe the evolution of the system' (p.10 l.58); do the authors mean that they decide to choose $D_{\rho\rho}(t)$ as variable of a simplified system?
 - (b) ' $D_{\rho\rho}(t)/\langle \rho(t) \rangle$ exhibits exponential behavior in the limit of vanishing variance' (p.11 l.39); what does it mean? from which equation do we see this fact?
 - (c) It seems that everything is actually based on Fig. 4 that shows, taking the averages over a number of simulations computed for some specific parameter values and initial conditions, that $D_{\rho\rho}(t)/\langle \rho(t) \rangle$ grows exponentially over time. Before jumping to a conclusion from that, I would like to know that this behaviour occurs for all initial conditions in some parameter region of $(\gamma/\alpha, N)$. It could also be useful if the two empirical parameters

(D_1 and τ) of this exponential behaviour depended on the parameters ($\gamma/\alpha, N$) and on the initial conditions in a predictable way.

5. The final question is what is the use of (16) or (17)? Can we learn something from these equations that was not known before?

In my opinion, the current manuscript cannot be published.

It is possible that a publication could be obtained from a thorough revision that includes appropriate references to the literature and focusses on what is actually novel. The main point should be showing how the analysis gives us more insight on the behaviour of the stochastic SIS model beyond what can already be gained by the existing literature or by simulating the model.

References

- [BP97] Bolker, B., & Pacala, S. W. (1997). Using moment equations to understand stochastically driven spatial patterns formation in ecological systems. *Theor. Pop. Biol.*, 52, 179-197.
- [KL89] Kryscio, R. J. & Lefevre, C. (1989) On the extinction of the S-I-S stochastic logistic epidemic. *J. Appl. Prob.* 27, 685-694.
- [K00] Keeling, M. J. (2000). Metapopulation moments: Coupling, stochasticity and persistence. *Journal of Animal Ecology*, 69(5), 725-736.
- [ME08] Allen, L.J.S. (2008) An introduction to stochastic epidemic models, in *Mathematical Epidemiology*, F.Brauer, P. van den Driessche & J.Wu (eds.), Springer, pp. 81-130.
- [N96] Nåsell, I. (1996). The Quasi-Stationary Distribution of the Closed Endemic SIS Model. *Advances in Applied Probability*, 28(3), 895-932.
- [N01] Nåsell, I. (2001) Extinction and Quasi-stationarity in the Verhulst Logistic Model, *J. theor. Biol.* 211, 11-27.
- [N03a] Nåsell, I. (2003). Moment closure and the stochastic logistic model. *Theoretical Population Biology*, 63(2), 159-168.
- [N03b] Nåsell, I. (2003) An extension of the moment closure method, *Theoretical Population Biology* 64, 233-239.

Appendix B

Dear Editor,

We have carefully read the Referee's report about the manuscript "Improved SIS epidemic equations based on uncertainties and autocorrelation functions". Their critiques are pertinent, and comments are well-advised. We sincerely thank them for their efforts and critical reading of the paper.

This new version of the manuscript captions much of the Referees' concerns, making the presentation more comprehensive. Our main point is that the fluctuations in epidemic processes cannot be neglected nor treated merely as a Langevin equation. The equations are obtained from first principles and validated by the literature. Once more, we thank the Referees for pointing many references, which we were not aware of. Furthermore, our presentation focus on the non-symmetric fluctuations, where our results pop up. The amendments are displayed in red in the revised manuscript.

We would like the Editor to consider this manuscript as a new submission since it differs significantly from the first one. Also, we believe the Referees will be glad to see all their concerns addressed, and that their suggestions contributed to a new version suitable to be published in Royal Society Open Science.

Cordially,

The Authors

Referee 1: The authors seem not to be very familiar with the literature on the topic, as can be seen by the lack of references to some of the main papers in which the model was studied, such as [KL89], [N96] and especially [N01]. Moreover, the use of correlation equations has been around for at least 25 years in the eco-epidemiological literature; for instance, equation (7) is (2) in [K00] (though in a different notation) and is at page 97 of the textbook [ME08]; the Gaussian approximation (9a)-(9b) is presented in Appendix A of [K00] while [N03a] and [N03b] are devoted to a systematic analysis of moment closure methods for the SIS stochastic model.

Reply: Acknowledged. We were unfamiliar with the works of Nasell and their implications on our study, including the implications of the absorbing state on the dynamics. Concerning the use of autocorrelation functions, the Referee is correct. The hyperbolic wording of the theme in the previous version of the manuscript leads to an inaccurate contextualization of the problem. To address these issues, we have expanded on the topics (Page 2 line 27) “More importantly, the stochastic analysis (...) to study time series of epidemiological data and assess the impact of spatial influences on stochastic fluctuations [15-20]”.

Additional citations have been included throughout the text for improved context. In particular, an expanded context of previous studies has been included in (Page 3 line 41) “(...) Numerical and analytical evidence (...) master equation of the disease spreading process”.

Referee 1: In several points in the manuscript (e.g. at page 4) the authors discuss how the equations for the moments can be used for extracting epidemiological parameters from data, as if one could have observations of $\langle \rho(t) \rangle$, or of the variance σ^2 . However $\langle \rho(t) \rangle$ is an ensemble average, and could be estimated only if we had a large number of independent realizations of the epidemic process, as in the simulations presented in Fig. 2. In reality, one has some observations over a single realization of the epidemic process (assuming that the model is correct), and inference must be performed from this kind of data. It is not clear how equations (7), (9) or (17) can contribute to this. Probably it is for this reason that moment equations (although well know for a long time) have rarely been considered for processes with homogeneous mixing, but rather used to approximate spatial epidemics, or so-called metapopulation models, where one assumes there is a large number of populations with weak interactions (see for instance [BP97]).

Reply: We understand the concern. In fact, that is the motivation to develop a second order differential equation for $\rho(t)$ instead of a system of differential equations for the average and variance. “ (...) the issue can be avoided entirely by combining the system of differential equations for $\rho(t)$ and $\sigma^2(t)$ into a single differential equation”.

The same rationale applies for autocorrelation functions. However, we acknowledge the previous version of the manuscript failed to explore the relationship between $\sigma^2(t)$ or $\langle \rho(t) \rangle$ with $D_{\rho\rho}(t)$. The revised manuscript improves the discussion on the relationship between the three variables in the non-Gaussian regime. (Page 14 line 235) “Eq. (12) agrees well with simulated data (...) For instance, numerical data suggests $\xi_{\text{est}} = 0.201$ (...)”. (Page 16 line 264)

From a methodological perspective, however, the underlying assumption of an ensemble (either from truly independent samples, or weakly interacting populations) is necessary to establish the nature of the uncertainties in the system. Moreover, it allows us to quantify it through simulations. Ideally, one would then define a scale for the noise, in close analogy with the temperature in statistical physics and other reaction-diffusion problems. This has not been done here.

Referee 1: The authors present the model as a continuous time Markov chain with 2^N states, corresponding to the N individuals. This may be necessary if contacts occur along some specific graphs; however, the authors analyse only the complete graph, in which individuals are indistinguishable, so that the process can be described on the state space $0, 1, \dots, N$, the number of infected individuals. This simplifies dramatically computations and notation (by the way, the authors could have made an effort to avoid the jargon of statistical physicists, and, when preparing a manuscript that should be widely read, to explain the notation used in that community).

Reply: We acknowledge the concern. In the early stages of the study, we considered more realistic contact networks. However, the uncertainties inherent to the dynamics and the one that steams from the network would interlace. By selecting only the complete graph, one can focus on dynamical fluctuations only, reducing the number of free parameters of the problem and a proper comparison with compartmental equations. (Page 5 line 103) “(...) using a stochastic agent-based approach to better grasp the emergence of uncertainties (...) external noise source to mimic fluctuations (Langevin formulation)”.

We expand on the subject (Page 6 line 125) “(...) The main advantage of using Eqs. (4) and (5) lies in their applicability for arbitrary networks (...) The choice also allows an adequate comparison with the compartmental equations”.

We also acknowledge the issues regarding jargons borrowed from statistical physics. In the revised version, some of them have been replaced. Furthermore, the notation is built entirely to facilitate the connections between agent-based simulations with algebraic and/or spectral methods. Such connections have been fruitful in the context of hermitian reaction-diffusion equations in systems with conformal invariance (Alcaraz et al, Ann. Phys. 1998).

Referee 1: The analysis of system (9) is incomplete. The authors are able to compute a class of explicit solutions (I am very surprised that the authors are able to find explicit solutions of a nonlinear second-order differential equation; honestly, I did not check that indeed (11) is a solution, suspecting that it must come from some general approach), but they miss the fact that these are not all the solutions. Indeed, a simple analysis of (9) (shown in my drawing below) shows that there is a saddle point at $E^* = (\rho_{\text{eq}}/2, \rho_{\text{eq}}^2/4)$. The stable manifold of E^* works as a separatrix: if initial conditions are below it, solutions asymptotically tend to $(\rho_{\text{eq}}, 0)$ and presumably are represented by (11); with initial conditions above the separatrix, $\langle \rho(t) \rangle$ becomes negative in finite time, so that the solutions lose biological realism. This fact was noted by [K00], who therefore suggested the use of multiplicative moments, based on lognormal approximation.

Reply: The Referee is correct. Our solution was incomplete as it could not capture one of the critical points on the separatrix $\sigma^2 = \langle \rho \rangle^2$. Although the proposed solution satisfies the differential equation, it is only valid below the separatrix. At the separatrix, the solution is slightly different as we show in the revised manuscript. Above the separatrix, we argue that the signal-to-noise ratio becomes small, in disagreement with the assumptions behind the equations of motion. To address these issues, we now explain how the solutions are obtained (inspired by projective transformations) as well as direction fields sketched by the Referee. (Page 17 line 281) “Recalling that (...) grows exponentially along time, producing negative solutions”.

- Referee 1:** The authors discuss in Section 3 Gaussian fluctuations, and in Section 4 the use of autocorrelation functions, but they are very vague about the use of either, though they say that Gaussian fluctuations should be inappropriate when $\gamma/\alpha \approx 1$ and when population is small. This probably corresponds to the parameter regions with qualitatively different behaviours much more clearly identified in [N01].
- Reply:** Indeed, that parameter region portrays the phenomenon discussed in the manuscript. However, it is small N rather than small γ/α that primarily dictates the influence of the absorbing state on the dynamics. (Page 10 line 180) “(...) However, $\Delta_3(t)$ also measures the fluctuation strength (...) fluctuations for fixed N : Gaussian and non-Gaussian fluctuations”. The topic is also discussed in the introduction, followed by the appropriate citations.
- Referee 1:** a) “Hence $D_{\rho\rho}(t)$ can be interpreted as an alternative metric to describe the evolution of the system” (p.10 1.58); do the authors mean that they decide to choose $D_{\rho\rho}(t)$ as variable of a simplified system?
- Reply:** Yes. For a noise-free system, $D_{\rho\rho}(t)$ would be equal to the relative variation of $\langle\rho(t)\rangle$. Its behavior changes drastically whether the fluctuations are able to drive the system to the absorbing state. Thus, it can be used to build the differential equation, and to assess the fluctuation regime. (Page 12 line 222) “In contrast, an exponential growth (...) fluctuations are in place”. And (Page 14 line 245) “For practical purposes, (...) as indicated in Fig. 8”.
- Referee 1:** b) “ $D_{\rho\rho}/\langle\rho(t)\rangle$ exhibits exponential behavior in the limit of vanishing variance” (p.11 1.39); what does it mean? from which equation do we see this fact?
- Reply:** It follows from Eq.(3) using the fact that for noise-free systems $D_{\rho\rho}(t)$ is the relative variation. (Page 12 line 214) “According to Eq. (3), an exponential decay of $D_{\rho\rho}(t)/\langle\rho(t)\rangle$ occurs whenever $\langle\rho(t)\rangle$ is reasonably described by compartmental equations.”
- Referee 1:** c) It seems that everything is actually based on Fig. 4 that shows, taking the averages over a number of simulations computed for some specific parameter values and initial conditions, that $D_{\rho\rho}(t)/\langle\rho(t)\rangle$ grows exponentially over time. Before jumping to a conclusion from that, I would like to know that this behaviour occurs for all initial conditions in some parameter region of $(\gamma/\alpha, N)$. It could also be useful if the two empirical parameters (D_1 and τ) of this exponential behaviour depended on the parameters $(\gamma/\alpha, N)$ and on the initial conditions in a predictable way.
- Reply:** The initial prediction for $|D_{\rho\rho}(t)/\langle\rho(t)\rangle|$ is based on the expected results for compartmental equation. In the non-Gaussian regime, one instead measures an exponential growth instead of constant value. We expand on the topic, including relation between the decay rates from $\langle\rho(t)\rangle$, $\sigma^2(t)$, and $D_{\rho\rho}(t)$ (summarized in Table I). Figure 4 and 7 show that the phenomenon is triggered for a particular threshold value γ/α for a fixed N . However, we have been unable to determine their exact relationship.
- Referee 1:** The final question is what is the use of (16) or (17)? Can we learn something from these equations that was not known before?
- Reply:** Acknowledged. As demonstrated by numerical simulations, if one were to favor compartmental equations over equations that included $D_{\rho\rho}(t)$, one would certainly obtain inaccurate predictions. Here, we suggest that instead, one should monitor, for example, $D_{\rho\rho}$ as a function of $\langle\rho(t)\rangle$ (or $D_{\rho\rho}(t)/\langle\rho(t)\rangle$). Not only the improved equations agree with numerical simulations, but also it would inhibit incorrect estimates of epidemiological parameters in small populations (see Figure 10). More specifically, an exponential decay for $\rho(t)$ does not necessarily mean $\gamma > \alpha$, due to fluctuations. (Page 14 line 235) “Eq. (12) agrees well with simulated data (...) numerical data suggests $\xi_{\text{est}} = 0.201$ (...)”.

-
- Referee 2:** First of all, I invite the authors to have a look at the work done in the paper titled “New Moment Closures Based on A Priori Distributions with Applications to Epidemic Dynamics” Bull Math Biol (2012) 74:1501–1515 DOI 10.1007/s11538-012-9723-3. In this paper the authors consider the SIS model on a fully connected network and derive ODEs for the first and second moments and propose the closure of the third moment in terms of the first and second. This is based on the assumption that $p_k(t)$ (the probability of observing k infected individuals at time “ t ”) is binomially distributed. They end up with a system of two ODEs. This is very similar to what is done in this new paper. They also show numerically that the difference between the exact system and the closed system seem to scale like $1/N^2$, which is an improvement over $1/N$ for some previously use closures. The normal distribution is also proposed as a potential candidate rather than the binomial. So I am not too sure that I see where the novelty of the paper lies and how it builds/adds/complements the state-of-the-art.
- Reply:** We agree. The study cited by the Referee and Vilar [Vilar and Rubi, Sci. Rep. 2018] produce the same equations under similar hypothesis. To address the issue we cite them explicitly and focus our findings primarily on the non-Gaussian regime. (Page 3 line 50) “The resulting differential equations for Gaussian fluctuations have been reported before [25, 26], and also derived in a more general formulation for population dynamics based on Langevin equations [27]”. And (Page 17 line 271) “ Both equations have been derived previously (see [26, 27])”.
- Referee 2:** There is little merit in simulating epidemics on graphs of size 50, since one has exact solutions by solving 51 linear ODEs, this can be done easily even for network where the number of nodes scale like $O(1000)$. This can be also used to compare the moments of the true model and that of the approximation.
- Reply:** We agree that the complete network oversimplifies simulations, reducing the effective degrees of freedom to $N + 1$. The simulations, however, are performed for comparison purposes with analytical results for small populations. We expand on the topic in (Page 2 line 39) “We find that uncertainties play an important role in small populations (...) epidemiological parameters from data”.
- Referee 2:** Since the whole analysis focuses on the fully connected network, there is not point in introducing overly complicated notation and talking about state space of size 2^N . In this case, the exact stochastic model is given by the forward Kolmogorov equations with $N+1$ states, and this is very well known. Even more well-known is Eq. (7) and this does no need to be derived, see of example Epidemic Modelling by Daley and Gani (done for SIR but is identical for SIS) or Mathematics of epidemics on networks by Kiss, Simon and Joel. Eq. (7) can simply be stated and referenced accordingly, or the paper cited at point (1).
- Reply:** We acknowledge the concern. The Dirac notation for vectors is ubiquitous in Physics but not necessarily true for other disciplines. It is reasonable that the entire section could be skipped now that we are aware of previous results. However, we ultimately decided to keep the section and notation: it allows researchers that use the same formalism to engage in the discussion, unveiling new spectral properties and generalizations. For instance, the review of reaction-diffusion processes under the lens of operators have guided to important findings. Among them, conformal invariance and connection with spin systems [Alcaraz et al, Ann. Phys. 1998]; or to obtain new equations for cancer growth [Delarouers et al, Phys. Rev. E 2009]. Furthermore, the step operator \hat{H} remains valid for arbitrary networks (Page 6 line 125) “The main advantage of using Eqs. (4) and (5) (...) reinforcing their validity for general networks”. Figure 2 has also been added to support the claim.

- Referee 2:** In the text below figure 1, the authors use reference [23] which seems to be their own, but again these facts are well known for some time, see also the second reference that I provided above or others. But again, I believe since the whole paper is based on the fully connected network, the very technical notation is not needed.
- Reply:** Acknowledged. See the previous reply.
- Referee 2:** Above equation (2) could the authors provide a reference for the extension of the α to α^* second/first moment? I am not sure this is totally correct.
- Reply:** Our statement was incorrect. The revised manuscript now reads (Page 4 line 79) “Explicit generalizations are available (...) of super-spreaders in real world spreading processes [3].”
- Referee 2:** Just above Section II (Complete Graph), the authors talk about simulation results but there is no mention how these are done, do they use the Gillespie algorithm or how are they done. I really would like to see comparisons between the average number of infected nodes taken from different model, rather than some transformed quantity.
- Reply:** The revised manuscript extends the section “Data accessibility”. It now includes, besides the source code and data, an explanation of the direct Monte Carlo simulation and its shared roots with the Gillespie algorithm. We also list the reasons to favor one to the other, for this particular research. (Page 4 line 94) “(see Data accessibility and Ref. [29] for further details)” ; and (Page 20 line 347) “Numerical simulations are performed (...) without additional processing algorithms or interpolation methods”.
- Referee 2:** The authors go on about and promote this new form of the equation but then they do not fit the model to any real-world epidemic, so why cast then the equation in an unfamiliar form if its usefulness is not shown.
- Reply:** We would argue there is merit in comparing our them with well-known established equations, emphasizing similarities and differences. Moreover, their usefulness is verified with simulations, even more so in regimes in which the classical equations fail.
- Referee 2:** In figure 2, has the Δ_3 been taken from simulation? Again, I would plot the average number of infected nodes and not only the approximations.
- Reply:** We used forward derivative for $\sigma^2(t)$ and $\Delta_3(t)$ (from simulated data) in Eq. (8), as shown in Figure 3 of the revised manuscript. Figure 4 has been added to showcase the effects of finite populations.
- Referee 2:** I could not get Eq 9b, is there a σ^2 missing on either the left- or right-hand side?
- Reply:** Eq. (9b) in the previous manuscript is now Eq. (15b) in the revised one. Note that derivative is applied to $\ln \sigma^2(t)$.
- Referee 2:** The Gaussian approximation only seem to work for some parameter values. Can the authors map this out, otherwise the model is not very useful as its range of operation is not known.
- Reply:** Unfortunately, we failed to derive an explicit relationship between γ/α and N , which would otherwise define the regime of the fluctuations. Instead, we propose one should monitor $D_{\rho\rho}(t)/\langle\rho(t)\rangle$ with t ; or $D_{\rho\rho}$ with $1/\langle\rho(t)\rangle$. (Page 14 line 245) “For practical purposes, one can either (...) as indicated in Fig. 8”
- Referee 2:** Effectively one needs to use two different models based on what values α and γ take.
- Reply:** The assessment is correct. We would add that it depends crucially on N for a given γ/α . One should use Eq. (16) for the Gaussian case. Eq. (13) is valid in general, but a good approximation for it is available in the non-Gaussian regime, via Eq. (14).

- Referee 2:** The stochastic version of this model is analysed in detail in the seminal book by Nasell (Extinction and quasistationarity in the stochastic Logistic SIS model).
- Reply:** We agree. Nasell works include many aspects of stochastic SIS model. The revised manuscript includes proper references to some of his works. (Page 3 line 44) “This issue has been examined in details before [21–24]”.
- Referee 2:** The authors motivate their work by taking about populations of small size yet in the derivation $N \gg 1$ everywhere. I am confused.
- Reply:** We inverted the section order to clarify the text. First, we explain the non-Gaussian regime and present their findings; afterward, equations for $\langle \rho(t) \rangle$ and $\sigma^2(t)$ are introduced for Gaussian fluctuations. $o(1/N)$ corrections are ignored only in the Gaussian section, producing a second order differential equation for $\langle \rho(t) \rangle$. The procedure frees the data collector from the burden of obtaining estimates for $\sigma^2(0)$.

Appendix C

Dear Editor,

After carefully reading the Referee's report, we now present our replies to their critiques and comments, as well as amendments in the new version of the manuscript "Improved SIS epidemic equations based on uncertainties and autocorrelation functions".

We want to thank Referee 1 for their concern about the usage of ensemble averages. The topic very often is dismissed and receives less attention than it deserves. At the same time, a detailed discussion of this problem is well beyond the scope of this manuscript. Readers who are not familiar with the issue will not be bogged down by the theoretical need of a perfect state of equilibrium for the determination of ensemble statistics. We believe all readers will appreciate that the averaging methodology and assumptions used in the paper are clearly spelled out. In the manuscript, we infer the effect of multiple realizations of the stochastic evolution from the evolution rules themselves, in a way reminiscent of the classical derivation of the diffusion equation from a random walker. The reader can decide whether the definitions we use will serve their purposes or is consistent with their hypothesis.

The main point of the paper remains: fluctuations in epidemic processes in small systems cannot be neglected, especially when fluctuations are non-symmetric. Amendments are displayed in red in the revised manuscript to make it easier to find in the text. The amendments and replies are detailed below. We hope that the new version of the manuscript adequately addresses the Referees' concerns and comments.

Cordially,

The Authors

Referee 1: I am still not convinced by their answer to my comment that ensemble averages are not observable quantities.

Reply: We agree that ensemble averages are not directly observable for a single instance of the stochastic evolution of a problem. At the same time, we know that equations describing the general behavior of stochastic variables can be inferred from ensemble averages. The best example is the random walker, whose ensemble averaged equation is the diffusion equation, with square displacement increasing linearly with time. For a single instance, however, the movement is erratic as if the particle were subjected to the action of a random force. Even in this scenario, the statistical properties of the random force are ultimately dictated in an ensemble average in order to make sense.

One way to mimic the ensemble averaging in a real-life epidemic situation is to partition the population into smaller subsets, and treat each subset as an instance of the ensemble. As long as each instance interacts only very weakly with each other, it can be a good way to build the ensemble. That is the idea behind statistical physics after Gibbs and Boltzmann, but also in metapopulations in biology and ecology. If the system is in equilibrium – which is not the case of the manuscript – one can also employ the ergodic theorem, ie, replace ensemble averages by averages over time.

The idealized ensemble, however, can still be used as starting points to produce equations for the averages and other statistics. In the revised manuscript, we expand on the topic in (Page 5-6 line 107-117): “We also note that (...) random walker.”

Referee 1: Anyway, I am not convinced of the usefulness of the authors’ approach to the problem, but others may think it otherwise. In that perspective, the computations presented in the manuscript may provide a useful contribution.

Reply: We acknowledge the comment. In this paper, we decided to consider a simple network first as we are still learning the caveats and implications of disease spreading in small systems. It was an important and necessary step to generalize it and to obtain the equivalent Fokker-Plank equations for statistical moments and correlations for arbitrary networks, or competing diseases (in preparation), or for growth of tumor cells with diffusive behavior.

Referee 1: I have only one small observation to the text: - at p.11 l.195-196 the authors write ”The effect can be found in small populations but it is enhanced in small populations:” There must be a typo.

Reply: The typo has been amended in the new version of the manuscript.

Referee 2:

The authors acknowledged my comments and made some changes as a result. Was disappointed that at least in two cases even though they agreed with my comments they did not implement them fully, see for example:

I said “There is little merit in simulating epidemics on graphs of size 50, since one has exact solutions by solving 51 linear ODEs, this can be done easily even for network where the number of nodes scale like $O(1000)$. This can be also used to compare the moments of the true model and that of the approximation.” The authors said: “We agree that the complete network oversimplifies simulations, reducing the effective degrees of freedom to $N + 1$. The simulations, however, are performed for comparison purposes with analytical results for small populations. We expand on the topic in (Page 2 line 39) “We find that uncertainties play an important role in small populations (...) epidemiological parameters from data”.”

Reply:

As we stated in the previous reply, we do agree that the problem becomes much simpler when the stochastic dynamics takes place in a complete network, because every node becomes, on average, equal to each other. We could have explored a far more complex network for the sake of ease of publication, but we wouldn't be able to compare our results with the well-known SIS compartmental model, and whether differences appear if fluctuations are taken into account (pag 7). Incidentally, fluctuations are more important in finite systems and for certain ranges of epidemiological parameters.

Concerning the second part – that there is little merit in simulating a small system as opposed to solving EDOs – we disagree. For four separate reasons.

First, the paper also deals with autocorrelation functions away from equilibrium (thus no ergodic theorem). Simulations provide a much easier way to probe autocorrelation functions. On top of that, simulations fit naturally to the description provided by Eq. (10).

Second, Eq. (4) explicitly shows that the equations for probabilities form a linear system. As the referee correctly points out, one can select the total number of infected as the relevant descriptor – it is a valid sector with dimension $N + 1$. As long as the initial state is in this particular sector, the referee's argument holds. For instance, a random infected person among a population of size N is a valid initial state. In contrast, an initial state in which a *specific* person is infected can only be formed by a linear combination of various eigenvectors, from different eigensectors, which in general is not described by only $N + 1$ equations. Of course, one can always neglect the additional corrections, but it would defeat the purpose of studying fluctuations that stems from finite size corrections.

Third, as far as simplifications go and if only the statistical moments in a complete network matter to the study, then there are even easier alternatives to the solution of $N + 1$ differential equations, with $N + 1$ initial conditions. Instead, one can simply define the generating function $G(x, t) = \sum_{k=0}^N P_k(t)x^k$, where $P_k(t)$ is the probability to measure k infected individuals at time t , with the additional constraints $P_{-1}(t) = P_{N+1}(t) = 0$. It is easy to show that the generating function satisfies $\partial_t G = (\gamma/x - \alpha)(1 - x)(x\partial_x)G + (\alpha/N)(1 - x)(x\partial_x)^2 G$. Because $\langle n^\ell \rangle = \lim_{x \rightarrow 1} (x\partial_x)^\ell G$, the equation for $G(x, t)$ gives access for the equations of motion for all statistical moments. The equation itself can be solved numerically, or can be put in a Sturm-Liouville form by assuming $G(x, t) = \sum_\mu c_\mu g_\mu(x) e^{-\lambda_\mu t}$, where c_μ are coefficients and λ_μ are eigenvalues – it is a hint from Eq. (4).

Fourth, even though operators in Eq. (5) may seem like a cannon to blow an ant, the formalism itself allows us to explore paths that can be useful for other situations in epidemiology or population growth. For instance, one can derive the space-time equations for the density in different networks, even in the presence of competing/coexisting diseases (in preparation), which often produces non-linear diffusion equations. It is important to note that these kinds of equations are far from intuitive and have important implications in tumor growth as well [Phys. Rev. E 79 031917 (2009)].

- Referee 2:** I said “Since the whole analysis focuses on the fully connected network, there is not point in introducing overly complicated notation and talking about state space of size $2N$”
The authors said: “We acknowledge the concern. The Dirac notation for vectors is ubiquitous in Physics but not necessarily true for other disciplines. It is reasonable that the entire section could be skipped now that we are aware of previous results. However, we ultimately decided to keep the section and notation:”
- Reply:** The Dirac notation for vectors and operators has been used by us and other researchers [e.g. Deroulers et al, Phys. Rev. E 79 031917 (2009)] because it makes state representation easy and compact.
- Referee 2:** It seems that there is still not test for how good the closed model is when compared to simulations in terms of the expected number of I nodes at time t .
- Reply:** As per advice of the referee in the previous report, there is little merit in spending too much time on the validity of Eqs. (15a) and (15b), as they have been established by Vilar and Rubi [Sci. Rep. 8 (2018)] in the limit $N \gg 1$.